# CAUSAL REPRESENTATION LEARNING IN TEMPORAL DATA VIA SINGLE-PARENT DECODING

## ABSTRACT

Scientific research often seeks to understand the causal structure underlying high-level variables in a system. For example, climate scientists study how phenomena, such as El Niño, affect other climate processes at remote locations across the globe. However, scientists typically collect low-level measurements, such as geographically distributed temperature readings. From these, one needs to learn both a mapping to causally-relevant latent variables, such as a high-level representation of the El Niño phenomenon and other processes, as well as the causal model over them. The challenge is that this task, called causal representation learning, is highly underdetermined from observational data alone, requiring other constraints during learning to resolve the indeterminacies. In this work, we consider a temporal model with a sparsity assumption, namely single-parent decoding: each observed low-level variable is only affected by a single latent variable. Such an assumption is reasonable in many scientific applications that require finding groups of low-level variables, such as extracting regions from geographically gridded measurement data in climate research or capturing brain regions from neural activity data. We demonstrate the identifiability of the resulting model and propose a differentiable method, *Causal Discovery with Single-parent Decoding* (CDSD), that simultaneously learns the underlying latents and a causal graph over them. We assess the validity of our theoretical results using simulated data and showcase the practical validity of our method in an application to real-world data from the climate science field.

## 1 INTRODUCTION

In scientific domains, we often seek to learn causal relationships between high-level variables. For example, climate scientists want to understand how major modes of climate variability, such as the El Niño Southern Oscillation (ENSO) affect weather patterns worldwide (76; 77; 62). Neuroscientists want to uncover how different brain regions may be defined and influence one another (69). Identifying true causal links in a network of correlations is challenging in itself, but to compound the difficulty, scientists typically collect low-level and noisy measurements in place of causally relevant high-level variables. For example, instead of recording the presence or absence of ENSO and its global impact, climate scientists measure sea-surface temperatures at many locations. Instead of measuring overall communication between brain regions, neuroscientists must work with proxy information such as blood flow or electrical activity in specific locations. Thus, scientific discovery requires causal representation learning: the coupled tasks of learning latent variables that represent semantically meaningful abstractions of the observed measurements and the quantification of causal relationships among these latents (81).

What makes causal representation learning particularly challenging from a theoretical perspective is the non-identifiability of the models: there are typically many solutions – mappings from observations to latents – that fit the observed measurements equally well. Of these many alternatives, only some disentangled solutions capture the semantics of the true latents while the other solutions entangle the latents, changing their semantics and making it impossible to then infer the causal relationships among the latents. As such, a key focus of causal representation learning is identifying the latents up to disentangled solutions using various inductive biases.

In this paper, we introduce a causal representation learning method for temporal observations, *Causal Discovery with Single-parent Decoding* (CDSD), a fully differentiable method that not only recovers disentangled latents, but also the causal graph over these latents. The assumption underlying CDSD,

that is crucial for identifiability, involves highly sparse mappings from latents to observations: each observed variable e.g., sea-level pressure at a given grid location on Earth, is a nonlinear function of a single latent variable. We call this *single-parent decoding*. While this condition is strong, such assumptions have given rise to interpretable latent variables models for gene expression (7), text (4), and brain imaging data (56). Although single-parent decoding may not fit the needs of some analyses (e.g., images), it leads to scientifically-meaningful groupings of observed variables for many scientific applications. For example, in climate science, the sparse mapping corresponds to latent spatial zones, each exhibiting similar weather patterns or trends in their climate.

A key innovation of this paper is that, with our sparse mapping assumption, we can identify the latents up to some benign indeterminacies (e.g., permutations) as well as the temporal causal graph over the latents. We prove these identifiability results theoretically, and verify empirically that they hold in simulated data. Furthermore, we demonstrate the practical relevance of our method and assumptions via an application to a real-world climate science task. Our results indicate that CDSD successfully partitions climate variables into geographical regions and proposes plausible *teleconnections* between them – remote interactions between distant climate or weather states (97) that have long been a target for climate scientists.

**Contributions.**

1. We propose a differentiable causal discovery approach that simultaneously learns both latent variables and a causal graph over the latents, based on time-series data. (Section 3)

2. We prove that the single-parent decoding assumption leads to the identifiability of both the latent representation and its causal graph. (Section 3.4, Proposition 2)

3. We evaluate our method both on synthetic data and a real-world climate science dataset in which relevant latents must be uncovered from measurements of sea-level pressure. (Section 4)

## 2    RELATED WORK

**Causal discovery from time-series data.**  Many causal discovery methods have been proposed for time-series data (77; 79). Constraint-based approaches, such as tsFCI (16), PCMCI+ (75) and TS-ICD (72), learn an equivalence class of directed acyclic graphs by iterative conditional independence testing. The proposed method is part of a line of work of score-based causal discovery methods that require a likelihood function to score each graph given the data. While standard score-based methods operate on a discrete search space of acyclic graphs (or Markov equivalence classes) that grows exponentially with the number of variables, continuous score-based methods enforce acyclicity only through a continuous acyclicity constraint, proposed by Zheng et al. (95). Some variants of these methods have been proposed specifically to handle time-series data with instantaneous connections (63; 86; 17). However, in contrast with CDSD, all the methods mentioned above do not address the problem of learning a latent representation.

**Causal representation learning.** Recently, the field of causal representation learning (81) has emerged with the goal of learning, from low-level data, representations that correspond to actionable quantities in a causal structure.[1] Since disentangling latent variables is impossible from independent and identically distributed samples (27; 51), existing works learn causal representations with weak supervision from paired samples (2; 8; 52; 89; 19), auxiliary labels (39; 40; 47; 26; 28; 25), and temporal observations (46; 50; 43; 92), or by imposing constraints on the map from latents to observations (57; 71; 96).

This paper fits into the last category of work on sparse decoding, which constrains each observed variable to be related to a sparse set of latent parents, either linearly (14; 56; 6; 44) or nonlinearly (57; 96; 71). In comparison, the *single-parent decoding* assumption that we use imposes a stronger form of sparsity, similar to some work on factor analysis (84; 56; 91; 45). In contrast, this paper develops an identifiable single-parent decoding model that is nonlinear and scales well with high-dimensional observations. The line of work on independent mechanism analysis (20; 70; 11) is also related to our identifiability result. The class of single-parent decoders we propose in this work is a subset of the class of decoders with Jacobians consisting of orthogonal columns. This work contributes to identification results in this category of research, a task which has proven to be challenging.

---

[1]As a side note, the general idea of aggregating several low-level observations in order to only consider causal relationships at a high level is somewhat reminiscent of causal discovery with typed variables (10), *causal abstractions* (73; 5) and *causal feature learning* (13) which was also applied to climate science (12).

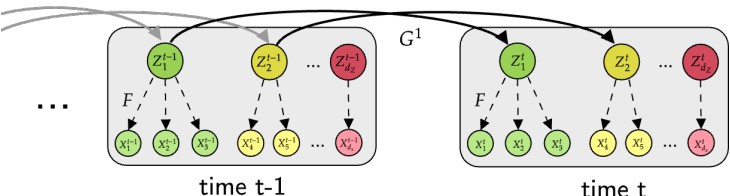

Figure 1: In the proposed generative model, the variables $z$ are latent and $x$ are observable variables. $G^k$ represents the connections between the latent variables, and $F$ the connections between the latents and the observables (dashed lines). The colors represent the different groups. For clarity, we illustrate here connections only up to $G^1$, but our method also leverages connections of higher order.

Finally, this paper also relates to Varimax-PCMCI (87), a method that, unlike causal representation learning, learns the latent variables and their causal graph in two separate stages. This method first applies *Principal Component Analysis* (PCA) and a Varimax rotation (33) to learn latent variables, as demonstrated in (62; 76), and then applies PCMCI (78), a temporal constraint-based causal discovery method, to recover the causal graph between the latents. In contrast, CDSD learns the latents and their temporal causal graph simultaneously via score-based structure learning, admitting nonlinearity in the relationships between latents as well as the mapping from latents to observations. Although Mapped-PCMCI supports nonlinear relationships between the latents, it does this via nonlinear conditional independence tests, which do not scale well (93; 85; 83). Nevertheless, we directly compare CDSD with Varimax-PCMCI in the experiments in Section 4.

## 3 CAUSAL DISCOVERY WITH SINGLE-PARENT DECODING

We consider the time series model illustrated in Fig. 1. We observe $d_x$-dimensional variables $\{x^t\}_{t=1}^T$ at $T$ time steps. The observed variables $x^t$ are a function of $d_z$-dimensional latent variables $z^t$. For example, the observations $x^t$ might represent temperature measurements at $d_x$ grid locations on Earth while the latents $z^t$ might correspond to unknown region-level temperature measurements.

We consider a stationary time series of order $\tau$ (i.e., $\tau$ is the maximum number of past observations that can affect the present observation) over the latent variables $z^1, \ldots, z^T$. Thus, we model the relationship between the latents at time $t$, $z^t$, and those at each of the $\tau$ previous time steps using binary matrices $\{G^k\}_{k=0}^{\tau}$ that represent causal graphs between the latent variables and their past states. That is, each matrix $G^k \in \{0,1\}^{d_z \times d_z}$ encodes the presence of lagged relations between the timestep $t-k$ and the present timestep $t$, i.e., $G_{ij}^k = 1$ if and only if $z_j^{t-k}$ is a causal parent of $z_i^t$. In what follows, we assume that there are no instantaneous causal relationships, i.e., the latents at time $t$ have no edges between one another in $G^0$ (see Appendix H.2 for a relaxation).

Finally, $F$ is the adjacency matrix of the bipartite causal graph with directed arrows from the latents $z$ to the variables $x$. We assume that $F$ has a specific structure: the *single-parent decoding* structure, where each variable $x_i$ has at most one latent parent. That is, the set of latent parents $z_{pa_i^F}$ of each $x_i$, where $pa_i^F$ is the set of indices of the parents in graph $F$, is such that $|pa_i^F| \leq 1$.

### 3.1 GENERATIVE MODEL

We now describe the model in detail that can be used to generate synthetic data.

**Transition model.** The transition model defines the relations between the latent variables $z$. We suppose that, at any given time step $t$, the latents are independent given their past:

$$p(z^t \mid z^{<t}) := \prod_{j=1}^{d_z} p(z_j^t \mid z^{<t}),$$ (1)

where the notation $z^{<t}$ is equivalent to $z^{t-1}, \ldots, z^{t-\tau}$. Each conditional is parameterized by a nonlinear function that depends on its parents:

$$p(z_j^t \mid z^{<t}) := h(z_j^t; \ g_j([G_{j:}^1 \odot z^{t-1}, \ldots, G_{j:}^\tau \odot z^{t-\tau}])),$$ (2)

where the bracket notation denotes the concatenation of vectors, $g_j$ denotes transition functions, $G_{j:}$ is the $j$-th row of the graph $G$, $\odot$ is the element-wise product, and $h$ is a density function of a continuous variable with support $\mathbb{R}$ parameterized by the outputs of $g_j$. In our experiments, $h$ is a Gaussian density although our identifiability result (Proposition 1) requires only that $h$ has full support.

**Observation model.** The observation model defines the relationship between the latent variables $\boldsymbol{z}$ and the observable variables $\boldsymbol{x}$. We assume conditional independence of the $x_j^t$:

$$p(\boldsymbol{x}^t \mid \boldsymbol{z}^t) := \prod_{j=1}^{d_x} p(x_j^t \mid \boldsymbol{z}_{pa_j^F}^t); \quad p(x_j^t \mid \boldsymbol{z}_{pa_j^F}^t) := \mathcal{N}(x_j^t; f_j(\boldsymbol{z}_{pa_j^F}^t), \sigma_j^2), \qquad (3)$$

where $f_j : \mathbb{R} \to \mathbb{R}$, and $\boldsymbol{\sigma}^2 \in \mathbb{R}_{>0}^{d_x}$ are decoding functions. As previously mentioned, we assume a specific structure of $F$, namely that $|pa_j^F| \leq 1$ for all nodes $x_j$. In the next section, we will present a way to enforce this structure.

**Joint distribution.** The complete density of the model is thus given by:

$$p(\boldsymbol{x}^{\leq T}, \boldsymbol{z}^{\leq T}) := \prod_{t=1}^{T} p(\boldsymbol{z}^t \mid \boldsymbol{z}^{<t}) p(\boldsymbol{x}^t \mid \boldsymbol{z}^t). \qquad (4)$$

## 3.2 EVIDENCE LOWER BOUND

The model can be fit by maximizing $p(\boldsymbol{x}^{\leq T}) = \int p(\boldsymbol{x}^{\leq T}, \boldsymbol{z}^{\leq T}) \, d\boldsymbol{z}^{\leq T}$, which unfortunately involves an intractable integral. Instead, we rely on variational inference and optimize an *evidence lower bound* (ELBO) for $p(\boldsymbol{x}^{\leq T})$, as is common to many instantiations of temporal *variational auto-encoders* (VAEs) (see Girin et al. (18) for a review).

We use $q(\boldsymbol{z}^{\leq T} \mid \boldsymbol{x}^{\leq T})$ as the variational approximation of the posterior $p(\boldsymbol{z}^{\leq T} \mid \boldsymbol{x}^{\leq T})$:

$$q(\boldsymbol{z}^{\leq T} \mid \boldsymbol{x}^{\leq T}) := \prod_{t=1}^{T} q(\boldsymbol{z}^t \mid \boldsymbol{x}^t); \quad q(\boldsymbol{z}^t \mid \boldsymbol{x}^t) := \mathcal{N}(\boldsymbol{z}^t; \tilde{\boldsymbol{f}}(\boldsymbol{x}^t), \operatorname{diag}(\tilde{\boldsymbol{\sigma}}^2)), \qquad (5)$$

where $\tilde{\boldsymbol{f}} : \mathbb{R}^{d_x} \to \mathbb{R}^{d_z}$ and $\tilde{\boldsymbol{\sigma}}^2 \in \mathbb{R}_{>0}^{d_z}$ are the encoding functions.

Using the approximate posterior and the generative model from Section 3.1, we get the ELBO:

$$\log p(\boldsymbol{x}^{\leq T}) \geq \sum_{t=1}^{T} \Big[ \mathbb{E}_{\boldsymbol{z}^t \sim q(\boldsymbol{z}^t \mid \boldsymbol{x}^t)} \big[ \log p(\boldsymbol{x}^t \mid \boldsymbol{z}^t) \big] - \mathbb{E}_{\boldsymbol{z}^{<t} \sim q(\boldsymbol{z}^{<t} \mid \boldsymbol{x}^{<t})} \mathrm{KL} \big[ q(\boldsymbol{z}^t \mid \boldsymbol{x}^t) \,\|\, p(\boldsymbol{z}^t \mid \boldsymbol{z}^{<t}) \big] \Big],$$
$$(6)$$

where KL stands for the Kullback-Leibler divergence. We show explicitly the derivation of this ELBO in Appendix A.

## 3.3 INFERENCE

We now present some implementation choices and our optimization problem of interest, namely maximizing the ELBO defined in Equation 6 with respect to the different parameters of our generative model.

**Latent-to-observable graph.** We parameterize $F$, the graph between the latent $\boldsymbol{z}$ and the observable $\boldsymbol{x}$, using a weighted adjacency matrix $W \in \mathbb{R}_{\geq 0}^{d_x \times d_z}$. Put formally, $W_{ij} > 0$ if and only if $x_i$ is a child of $z_j$. In order to enforce the single-parent decoding assumption for $F$, we follow Monti and Hyvärinen (56) and constrain $W$ to be non-negative and have columns that are orthonormal vectors.[2] From these constraints our single-parent decoding assumption follows: at most one entry per row of $W$ can be non-zero, i.e., a given $x_i$ can have at most one parent. As stated earlier, these constraints on $W$ are essential since they ensure that $W$ is identifiable up to permutation (we elaborate on identifiability in Section 3.4).

**Encoding/decoding functions.** We parameterize the decoding functions $f_j$ in Equation 3 with a neural network $r_j$ whose input is filtered using $W$ as a mask: $f_j(\boldsymbol{z}_{pa_j^F}^t) = r_j(W_{j:}\boldsymbol{z}^t)$. In Appendix E

---

[2]Note that, to simplify, we will sometimes say that $W$ is *orthogonal* even if it is not a square matrix; by that, we specifically mean that its columns are orthonormal vectors.

we show an architecture for the functions $r_j$ that leverages parameter sharing using only one neural network. For all experiments in the linear setting, we take $r_j$ to be the identity function as in Monti and Hyvärinen (56). The encoding function $\tilde{\boldsymbol{f}}$ (Equation 5) and the functions $g_j$ from the transition model (Equation 2) are also parameterized using neural networks.

**Continuous optimization.** We use $\phi$ to denote the parameters of all neural networks $(r_j, g_j, \tilde{\boldsymbol{f}})$ and the learnable variance terms at Equations 3 and 5. To learn the graphs $G^k$ via continuous optimization, we use a similar approach to Ke et al. (36); Brouillard et al. (9); Ng et al. (59), where the graphs are sampled from distributions parameterized by $\Gamma^k \in \mathbb{R}^{d_z \times d_z}$ that are learnable parameters. Specifically, we use $G_{ij}^k \sim Bernoulli(\sigma(\Gamma_{ij}^k))$, where $\sigma(\cdot)$ is the sigmoid function. To simplify the notation, we use $G$ and $\Gamma$ as the sets $\{G^1, \ldots, G^\tau\}$ and $\{\Gamma^1, \ldots, \Gamma^\tau\}$ in the remainder of the presentation. This results in the following constrained optimization problem:

$$\max_{W, \Gamma, \phi} \mathbb{E}_{G \sim \sigma(\Gamma)} \big[\mathbb{E}_{\boldsymbol{x}} \left[\mathcal{L}_{\boldsymbol{x}}(W, \Gamma, \phi)\right]\big] - \lambda_s ||\sigma(\Gamma)||_1$$
$$\text{s.t. } W \text{ is orthogonal and non-negative,}$$
(7)

where $\mathcal{L}_{\boldsymbol{x}}$ is the ELBO corresponding to the right-hand side term in Equation 6 and $\lambda_s > 0$ is a coefficient for the regularisation of the graph sparsity. To enforce the non-negativity of $W$, we use the projected gradient on $\mathbb{R}_{\geq 0}$ (see Appendix C.2). As for the orthogonality of $W$, we enforce it using the following constraint:

$$h(W) := W^T W - I_{d_z} .$$

We relax the constrained optimization problem by using the *augmented Lagrangian method* (ALM), which amounts to adding a penalty term to the objective and incrementally increasing its weight during training ((60); see Appendix C.1). Hence, the final optimization problem is:

$$\max_{W, \Gamma, \phi} \mathbb{E}_{G \sim \sigma(\Gamma)} \big[\mathbb{E}_{\boldsymbol{x}}[\mathcal{L}_{\boldsymbol{x}}(W, \Gamma, \phi)]\big] - \lambda_s ||\sigma(\Gamma)||_1 - \text{Tr}\left(\lambda_W^T h(W)\right) - \frac{\mu_W}{2} ||h(W)||_2^2, \quad (8)$$

where $\lambda_W \in \mathbb{R}^{d_z \times d_z}$ and $\mu_W \in \mathbb{R}_{>0}$ are the coefficients of the ALM.

We use stochastic gradient descent to optimize this objective. To estimate the gradients w.r.t. the parameters $\Gamma$, we use the Straight-Through Gumbel estimator (53; 30). In the forward pass, we sample $G$ from the Bernoulli distributions, while in the backward pass, we use the Gumbel-Softmax samples. This estimator was successfully used in several causal discovery methods (34; 9; 59). For the ELBO optimization, we follow the classical VAE models (41) by using the reparametrization trick and a closed-form expression for the KL divergence term since both $q(\boldsymbol{z}^t \mid \boldsymbol{x}^t)$ and $p(\boldsymbol{z}^t \mid \boldsymbol{z}^{<t})$ are multivariate Gaussians. Using these tricks we can learn the graphs $G$ and the matrix $W$ end-to-end. For a more detailed exposition of the implementation, such as the neural network's architecture, see Appendix E. For an extension of this approach to support instantaneous relations, see Appendix H.2.

### 3.4 IDENTIFIABILITY ANALYSIS

In this section, we discuss the identifiability of the model specified in Section 3.1. Put informally, we show that any solution that fits the ground-truth exactly recovers the true latents $\boldsymbol{z}^t$ up to permutation and coordinate-wise transformations, i.e., transformations that preserve the semantics of the latents and admit valid causal discovery.

To formalize identifiability, we first state an important result that allows to show that two models expressing the same distribution over observations must have i) the same decoder image and ii) the relationship between latent representations must be a diffeomorphism. Similar results have been show in previous literature (38; 46; 42; 1), and for completeness' sake we state it and its proof in Appendix B. For conciseness, we use $\boldsymbol{f}, \boldsymbol{g}$ as the concatenations of functions $[f_1, \ldots, f_{d_x}]$ and $[g_1, \ldots, g_{d_z}]$.

**Proposition 1** (Identifiability of $\boldsymbol{f}$ and $p(\boldsymbol{z}^{\leq T})$ up to diffeomorphism)**.** *Assume we have two models $p(\boldsymbol{x}^{\leq T}, \boldsymbol{z}^{\leq T})$ and $\hat{p}(\boldsymbol{x}^{\leq T}, \hat{\boldsymbol{z}}^{\leq T})$ as specified in Section 3.1 with parameters $(\boldsymbol{g}, \boldsymbol{f}, G, \sigma^2)$ and $(\hat{\boldsymbol{g}}, \hat{\boldsymbol{f}}, \hat{G}, \hat{\sigma}^2)$, respectively. Assume further that $\boldsymbol{f}$ and $\hat{\boldsymbol{f}}$ are diffeomorphisms onto their respective images and that $d_z < d_x$ (we do not assume single-parent decoding). Therefore, whenever $\int p(\boldsymbol{x}^{\leq T}, \boldsymbol{z}^{\leq T}) d\boldsymbol{z}^{\leq T} = \int \hat{p}(\boldsymbol{x}^{\leq T}, \hat{\boldsymbol{z}}^{\leq T}) d\hat{\boldsymbol{z}}^{\leq T}$ for all $\boldsymbol{x}^{\leq T}$, we have $\boldsymbol{f}(\mathbb{R}^{d_z}) = \hat{\boldsymbol{f}}(\mathbb{R}^{d_z})$ and*

$\boldsymbol{v} := \boldsymbol{f}^{-1} \circ \hat{\boldsymbol{f}}$ *is a diffeomorphism. Moreover, the density of the ground-truth latents* $p(\boldsymbol{z}^{\leq T})$ *and the density of the learned latents* $\hat{p}(\hat{\boldsymbol{z}}^{\leq T})$ *are related via*

$$p(\boldsymbol{v}(\hat{\boldsymbol{z}}^{\leq T})) \prod_{t=1}^{T} |\det D\boldsymbol{v}(\hat{\boldsymbol{z}}^t)| = \hat{p}(\hat{\boldsymbol{z}}^{\leq T}), \; \forall \hat{\boldsymbol{z}}^{\leq T} \in \mathbb{R}^{d_z \times T} \, .$$

The following paragraphs discuss how the structure in $F$ can be leveraged to show that $\boldsymbol{v}$ must be a trivial indeterminacy like a permutation composed with element-wise transformations.

**Identifiability via the single-parent structure of** $F$**.** The following proposition can be combined with Proposition 1 to show that the model specified in Section 3.1 with the single-parent decoding structure has a representation that is identifiable up to permutation and element-wise invertible transformations. The proof of this result can be found in Appendix B.

**Proposition 2** (Identifying latents of $\boldsymbol{f}$). *Let* $\boldsymbol{f} : \mathbb{R}^{d_z} \rightarrow \mathbb{R}^{d_x}$ *and* $\hat{\boldsymbol{f}} : \mathbb{R}^{d_z} \rightarrow \mathbb{R}^{d_x}$ *be two diffeomorphisms onto their image* $\boldsymbol{f}(\mathbb{R}^{d_z}) = \hat{\boldsymbol{f}}(\mathbb{R}^{d_z})$. *Assume both* $\boldsymbol{f}$ *and* $\hat{\boldsymbol{f}}$ *have a single-parent decoding structure, i.e.* $|pa_j^F| \leq 1$ *and* $|pa_j^{\hat{F}}| \leq 1$. *Then, the map* $\boldsymbol{v} := \boldsymbol{f}^{-1} \circ \hat{\boldsymbol{f}}$ *has the following property: there exists a permutation* $\pi$ *such that, for all* $i$, *the function* $\boldsymbol{v}_i(\boldsymbol{z})$ *depends only on* $\boldsymbol{z}_{\pi(i)}$.

Can we identify the causal graph $G$ over latent variables? The above result, combined with Proposition 1, shows that we can identify the distribution $p(\boldsymbol{z}^{\leq T})$ up to permutation and trivial reparameterizations. The question then reduces to "can we identify the causal graph $G$ from $p(\boldsymbol{z}^{\leq T})$?", which is the central question of causal discovery. It is well-known that, in the absence of instantaneous causal connections, a temporal causal graph can be identified from the observational distribution (65, Theorem 10.1). It is thus possible, without instantaneous connections, to identify $G$ (up to permutation) from $p(\boldsymbol{x}^{\leq T})$.

## 4 EXPERIMENTS

We empirically study the performance of CDSD on a number of linear and nonlinear settings using synthetic datasets. First, we compare CDSD to Varimax-PCMCI (87), an alternate method that has a closely similar application. The results, reported at Section 4.1, emphasize the advantages of CDSD over Varimax-PCMCI, especially in nonlinear settings. Next, we show that CDSD also compares favorably to identifiable representation methods (iVAE (39) and DMS (46)) on synthetic data that respect the single-parent decoding assumption. Finally, we apply CDSD to a real-world climate science task and show that it recovers spatial aggregations related to known climate phenomena, such as the El Niño Southern Oscillation (Section 4.2).

An implementation of CDSD is available at https://anonymous.4open.science/r/d4ty-F58B. For Varimax-PCMCI, we follow the implementation of Tibau et al. (87), where dimensionality reduction is done by combining PCA with a Varimax rotation (33), the causal graph is learned with PCMCI+ (75), and conditional independence is tested using a partial correlation test when latent dynamics are linear or the CMI-knn test (74) otherwise. Note that while PCMCI+ supports instantaneous connections, we always restrict the minimum time lag considered to 1. For further implementation details on both methods, refer to Appendix E.

### 4.1 SYNTHETIC DATA BENCHMARK

The first task is to compare CDSD and Varimax-PCMCI. The key modeling components to evaluate in the two compared methods are: linear versus nonlinear dynamics in the learned causal graphs over latents, and linear versus nonlinear decoding functions from latents to observations. CDSD can flexibly handle all these settings, whereas Varimax-PCMCI assumes linear maps from latents to observations. We evaluate the methods in the following cases: 1) linear dynamics and linear decoding, 2) nonlinear dynamics and linear decoding, and 3) linear dynamics and nonlinear decoding. We expect to find that CDSD shows clear advantages when the mappings from latents to observations are nonlinear. We also compare to other causal representation methods to show the identifiability gain induced by using the constraints on $W$ for data respecting the single-parent decoding assumption.

**Datasets.** We consider datasets randomly generated according to the model described at Section 3.1. The generative process is described in detail in Appendix D.1. Unless otherwise specified, we consider

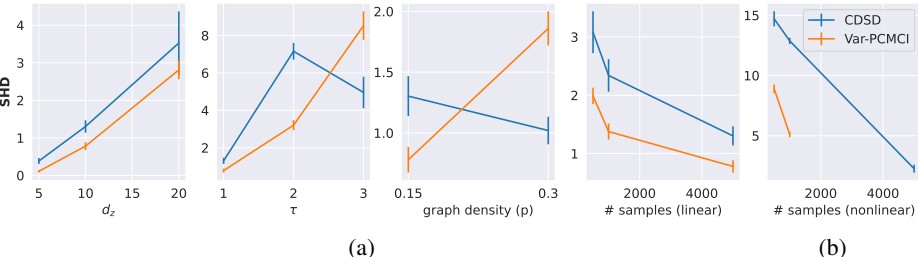

Figure 2: Comparison of Varimax-PCMCI and CDSD in terms SHD (lower is better) on simulated datasets with linear decoding and both a) linear and b) nonlinear latent dynamics.

$T = 5000$ timesteps, a stationary process of order $\tau = 1$, $d_x = 100$ observed variables, $d_z = 10$ latent variables, and random latent dynamic graphs, akin to Erdős-Rényi graphs, with a probability $p = 0.15$ of including an edge. The nature of the relationships among latents – and from latents to observables – is either linear or nonlinear, depending on the specific experiment.

**Protocol.** We assess variability in each experimental condition by repeating each experiment 100 times with different randomly generated datasets. The hyperparameters of both methods are chosen to maximize overall performance on 10 randomly generated datasets distinct from the evaluation (see Appendix F). Note that, for both methods, $d_z$ and $\tau$ are not part of the hyperparameter search and are set to the ground-truth values in the generative process.

**Metrics.** Performance is assessed using two metrics: i) mean correlation coefficient (MCC), which measures the quality of the learned latent representation, and ii) structural Hamming distance (SHD), which measures the number of incorrect edges in the learned causal graph. MCC corresponds to the highest correlation coefficient between the estimated latents ($\hat{z}$) and the ground-truth latent ($z$) across all possible permutations (as described in (40)). The use of permutations is necessary since identification can only be guaranteed up to a permutation (see Section 3.4).

**1) Linear latent dynamics, Linear decoding.** We start by evaluating the methods in a context where all causal relationships are linear. We consider a variety of conditions: $d_z = \{5, 10, 20\}$, $\tau = \{1, 2, 3\}$, $T = \{500, 1000, 5000\}$, and $p = \{0.15, 0.3\}$ (which corresponds to sparse and dense graphs). We observed that both methods achieve a high MCC $\geq 0.95$, in all conditions, which is not surprising since they are both capable of identifying the latents when the decoding function is linear (see Appendix G.2). The average SHD and its standard error are reported at Fig. 2a. Varimax-PCMCI performs slightly better than CDSD in most conditions, except for more challenging cases such as stationary processes of greater order ($\tau = 3$) and denser graphs ($p = 0.3$). The latter result is in line with previous studies, which observed that continuous optimization methods tend to outperform their constraint-based counterparts (95; 9) in dense graphs.

**2) Nonlinear latent dynamics, Linear decoding.** We now consider the case where causal relationships between the latents are nonlinear, while those from latents to observables remain linear. The results are reported at Fig. 2b. In contrast with the linear case, we do not present the results under all the experimental conditions due to the prohibitive running time of Varimax-PCMCI, which was greater than 24 hours for a single experiment (for the complete results, see Appendix G.2). This can be explained by its reliance on nonlinear conditional independence tests whose running time scales unfavorably w.r.t. the number of samples and variables (74; 94; 85). Consequently, results for Varimax-PCMCI are only reported up to 1000 samples. In sharp contrast, CDSD completed all experiments in a timely manner. Hence, while its solutions tend to have slightly higher SHD, CDSD can be used in contexts where Varimax-PCMCI, at least with a non-parametric conditional independence test, cannot.

**3) Linear latent dynamics, Nonlinear decoding.** The purpose of this experiment is to showcase the inability of Varimax-PCMCI to identify the latent representation when the relationships between the latents and the observables are nonlinear. This is the case, since PCA with a Varimax rotation is a linear dimension-reduction method. In contrast, CDSD should have no problem identifying the latents in this setting. We consider a dataset generated with the previously stated default conditions, where we ensure that the identifiability conditions of Section 3.4 are satisfied. The results are reported at Fig. 3. As expected, Varimax-PCMCI fails to recover the latent representation, achieving a poor MCC

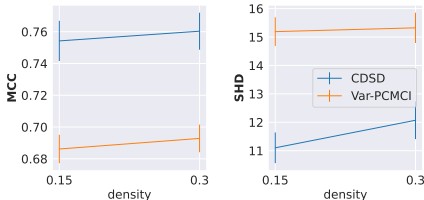
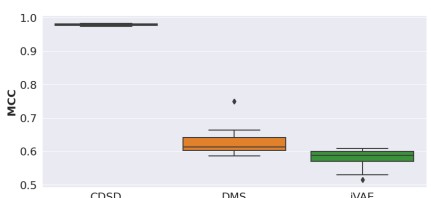

Figure 3: Comparison of CDSD and Varimax-PCMCI in terms of MCC (higher is better) and SHD (lower is better) on simulated datasets with linear dynamics and nonlinear decoding.

Figure 4: Comparison of CDSD to DMS and iVAE in terms of MCC.

and, consequently, a poor SHD. In contrast, CDSD performs much better according to both metrics. These results clearly show the superiority of CDSD over Varimax-PCMCI when the relationships between latents and observables are nonlinear because of the linearity assumption in the Varimax step.

**Comparison to causal representation methods.** We compare CDSD to two causal representation methods, iVAE (40) and DMS (46), on the synthetics data sets with linear decoding and nonlinear dynamics in Figure 4. For a fair comparison, we implement iVAE and DMS by modifying the objective for method in each case. For iVAE, this corresponds to not applying the constraints on $W$, not using regularisation on the graph $G$ (i.e., $\lambda_s = 0$) and fitting the variance of $z^t | z^{t-1}$. For DMS, it simply corresponds to not applying the constraints on $W$. Both methods have a worse MCC than CDSD. This is in line with our theoretical result since only CDSD leverages the single-parent decoding assumption. Note however that several assumptions required by iVAE and DMS may not hold in our datasets. For example, the identifiability result of iVAE assumes that the variance of $z^t | z^{t-1}$ varies sufficiently, which is not the case in our synthetic data. For DMS, while we use sparse transition graphs ($G$), we did not verify if the graphical criterion required by Lachapelle et al. (46) is respected (Theorem 5, assumption 5), nor whether its assumptions of sufficient variability hold. In Appendix I.3, we conduct a similar ablation for Varimax-PCMCI, which shows the necessity of the Varimax rotation in order for PCA to recover a good latent representation for the case of linear latent dynamics.

## 4.2 REAL-WORLD APPLICATION TO CLIMATE SCIENCE

To test the capabilities of CDSD in real-world settings, we apply it to the National Oceanic and Atmospheric Administration's (NOAA) *Reanalysis 1 mean sea-level pressure (MSLP)* dataset (35). Local variations in MSLP reflect changes in the dynamical state of the atmosphere for a certain region or, in other words, the occurrence and passing of weather systems (e.g. low- or high-pressure systems). Over time, MSLP data can thus be used to identify regions that share common weather properties, and to understand how regional weather systems are coupled to each other globally. Identifying such causal relationships would help climate scientists to better understand Earth's global dynamical weather system and could provide leverage for data-driven forecasting systems.

Here, we use MSLP data from 1948–2022 on a global regular grid with a resolution of $2.5°$ longitude $\times$ $2.5°$ latitude. We aggregated the daily time-series to weekly data and regridded it onto an icosahedral-hexagonal grid (see Appendix D.2) (55). The resulting dimensions are $3900 \times 6250$, covering 52 weeks of 75 years ($T = 3900$) and $d_x = 6250$ grid cells. We apply CDSD in order to cluster regions of similar weather properties, and identify which regions are causally linked to weather phenomena in other regions. We use the method with linear dynamics and linear decoding, and, similarly to (76), we use $d_z = 50$ and $\tau = 5$.

Fig. 5a shows the learned spatial aggregations and the causal graph $G$ obtained with CDSD. The learned aggregations match well with the coarser climatological regions used in the latest climate change assessment reports of the Intergovernmental Panel on Climate Change (IPCC), which were manually defined (compare to Figure 1 in (29)). The learned clusters broadly reflect a superposition of the effects of transport timescales, ocean-to-land boundaries, and the zonal and meridional patterns of the tropospheric circulation (Hadley, Ferrel, and polar cells) in both hemispheres. Among the visually most prominent features is the identification of East Pacific, Central Pacific, and Western Pacific clusters (clusters 13, 39, 48) along the tropical Pacific. These zones are well-known to be coupled through ENSO,

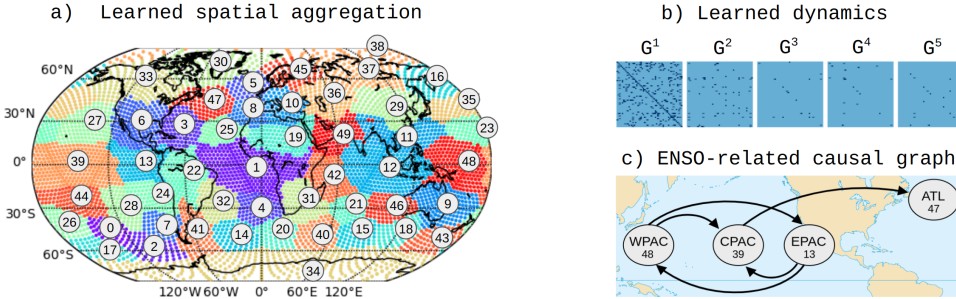

Figure 5: Overview of the climate science results for CDSD. a) Segmentation of the Earth's surface according to $W$. The groups are colored and numbered based on the latent variable to which they are related. b) Adjacency matrices for latent dynamic graphs $G^1, \ldots, G^5$, shown as $d_z \times d_z$ heatmaps. c) Subgraph of $G^1$ showing the learned causal relationships between known ENSO-related regions.

but the East Pacific typically sees the most pronounced temperature oscillations due to its shallow oceanic thermocline (e.g., 61). We also recover a relatively zonal structure of clusters in the Southern Hemisphere mid-latitudes (clusters 17, 2, 41, 14, 20, 40, 15, 18, and 43 from west to east) where the zonal tropospheric circulation moves relatively freely without significant disturbances from land boundaries. While not strictly enforced, all the learned regions are spatially connected/homogeneous, i.e. not divided into several de-localized parts (see Appendix G.3). Highly de-localized, globally distributed, components are for example a major issue in interpreting standard principal component analyses of MSLP data (21). In contrast, the regions learned by CDSD without constraints are not localized and are harder to associate to known regions such as those related to ENSO (see Appendix G.3).

While a detailed analysis of the learned causal graphs (Fig. 5b) is beyond the scope of this study, it is intuitive that the strongest and most frequent connections are found within a timescale of one week ($G^1$), but notably longer - likely more distant connections - are found, too. These likely reflect the well-known presence of long-distance teleconnections between world regions (62). In Fig. 5c we show one example of the causal coupling inferred for ENSO-related modes (clusters 13, 39, 47, 48) which is similar to the causal graph found in Runge et al. (78).

## 5    DISCUSSION

We present CDSD, a method that relies on the single-parent decoding assumption to learn a causal representation and its connectivity from time series data. The method is accompanied by theoretical results that guarantee the identifiability of the representation up to benign transformations. The key benefits of CDSD over Varimax-PCMCI are that i) it supports nonlinear decoding functions, and ii) as opposed to its constraint-based tests, it scales well in the number of samples and variables in the nonlinear dynamics case. Furthermore, as illustrated in the application to climate science, CDSD and its assumptions, appear to be applicable in practice and seem particularly well-suited for problems of scientific interest, such as the spatial clustering of weather measurements.

We highlight a few limitations that should be considered. Several assumptions, such as the stationarity of the dynamical system or the single-parent decoding assumption, can be partially or totally violated in real-world applications. We did not study the impact of these model misspecifications on the performance of CDSD. In all our experiments, we assumed that $d_z$ and $\tau$ were known. However, in practical applications, these values are unknown and might be difficult to infer, even for experts.

Besides these limitations, CDSD, in its general form, can be used in several contexts or be readily extended. It can be used with multivariate data (e.g., in climate science applications, one could be interested in modeling sea-level pressure, but also temperature, precipitation, etc.). Furthermore, in other contexts, such as brain imaging studies, one could be interested in learning different graphs $G$ for different subjects, while sharing a common spatial aggregation (as in Monti and Hyvärinen (56)). We want to highlight that the method can be further extended to include instantaneous connections, and learn from observational and interventional data. In Appendix H.1, we show how our method can be adapted to support all these cases. Overall, we believe that CDSD is a significant step in towards the goal of bridging the gap between causal representation learning and scientific applications.

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

## A    DERIVATION OF THE ELBO

We show explicitly the derivation of the ELBO by starting from the marginal likelihood and by using the model we have proposed in Section 3.1.

$$\log p(\boldsymbol{x}^{\leq T}) = \mathbb{E}_{\boldsymbol{z}^{\leq T} \sim q(\boldsymbol{z}^{\leq T} | \boldsymbol{x}^{\leq T})} \left[ \log p(\boldsymbol{x}^{\leq T}) \frac{p(\boldsymbol{x}^{\leq T}, \boldsymbol{z}^{\leq T})}{p(\boldsymbol{x}^{\leq T}, \boldsymbol{z}^{\leq T})} \frac{q(\boldsymbol{z}^{\leq T} | \boldsymbol{x}^{\leq T})}{q(\boldsymbol{z}^{\leq T} | \boldsymbol{x}^{\leq T})} \right] \tag{9}$$

$$= \mathbb{E}_{\boldsymbol{z}^{\leq T} \sim q(\boldsymbol{z}^{\leq T} | \boldsymbol{x}^{\leq T})} \left[ \log \frac{q(\boldsymbol{z}^{\leq T} | \boldsymbol{x}^{\leq T})}{p(\boldsymbol{z}^{\leq T} | \boldsymbol{x}^{\leq T})} + \log \frac{p(\boldsymbol{x}^{\leq T}, \boldsymbol{z}^{\leq T})}{q(\boldsymbol{z}^{\leq T} | \boldsymbol{x}^{\leq T})} \right] \tag{10}$$

$$= \text{KL}(q(\boldsymbol{z}^{\leq T} | \boldsymbol{x}^{\leq T}) || p(\boldsymbol{z}^{\leq T} | \boldsymbol{x}^{\leq T})) + \mathbb{E}_{\boldsymbol{z}^{\leq T} \sim q(\boldsymbol{z}^{\leq T} | \boldsymbol{x}^{\leq T})} \left[ \log \frac{p(\boldsymbol{x}^{\leq T}, \boldsymbol{z}^{\leq T})}{q(\boldsymbol{z}^{\leq T} | \boldsymbol{x}^{\leq T})} \right]. \tag{11}$$

Since KL $\geq 0$, we have:

$$\log p(\boldsymbol{x}^{\leq T}) \geq \mathbb{E}_{\boldsymbol{z}^{\leq T} \sim q(\boldsymbol{z}^{\leq T} | \boldsymbol{x}^{\leq T})} \left[ \log \frac{p(\boldsymbol{x}^{\leq T}, \boldsymbol{z}^{\leq T})}{q(\boldsymbol{z}^{\leq T} | \boldsymbol{x}^{\leq T})} \right]. \tag{12}$$

Now we will replace $p(\boldsymbol{x}^{\leq T}, \boldsymbol{z}^{\leq T})$ and $q(\boldsymbol{z}^{\leq T} | \boldsymbol{x}^{\leq T})$ by the factorisation we assumed in Equation 4 and 5:

$$\log p(\boldsymbol{x}^{\leq T}) \geq \mathbb{E}_{\boldsymbol{z}^{\leq T} \sim q(\boldsymbol{z}^{\leq T} | \boldsymbol{x}^{\leq T})} \left[ \log \frac{\prod_{t=1}^{T} p(\boldsymbol{z}^t | \boldsymbol{z}^{<t}) p(\boldsymbol{x}^t | \boldsymbol{z}^t)}{\prod_{t=1}^{T} q(\boldsymbol{z}^t | \boldsymbol{x}^t)} \right] \tag{13}$$

$$\geq \mathbb{E}_{\boldsymbol{z}^{\leq T} \sim q(\boldsymbol{z}^{\leq T} | \boldsymbol{x}^{\leq T})} \left[ \sum_{t=1}^{T} \log p(\boldsymbol{x}^t | \boldsymbol{z}^t) \right] + \mathbb{E}_{\boldsymbol{z}^{\leq T} \sim q(\boldsymbol{z}^{\leq T} | \boldsymbol{x}^{\leq T})} \left[ \sum_{t=1}^{T} \log \frac{p(\boldsymbol{z}^t | \boldsymbol{z}^{<t})}{q(\boldsymbol{z}^t | \boldsymbol{x}^t)} \right]. \tag{14}$$

Finally, thanks to the decomposition of our proposed posterior (Equation 5), we have:

$$\log p(\boldsymbol{x}^{\leq T}) \geq \sum_{t=1}^{T} \Big[ \mathbb{E}_{\boldsymbol{z}^t \sim q(\boldsymbol{z}^t | \boldsymbol{x}^t)} \left[ \log p(\boldsymbol{x}^t | \boldsymbol{z}^t) \right] - \tag{15}$$

$$\mathbb{E}_{\boldsymbol{z}^{<t} \sim q(\boldsymbol{z}^{<t} | \boldsymbol{x}^{<t})} \text{KL} \left[ q(\boldsymbol{z}^t | \boldsymbol{x}^t) || p(\boldsymbol{z}^t | \boldsymbol{z}^{<t}) \right] \Big]. \tag{16}$$

## B    IDENTIFIABILITY

In what follows, we overload the notation by defining $\boldsymbol{f}(\boldsymbol{z}^{\leq T}) := [\boldsymbol{f}(\boldsymbol{z}^1) \dots \boldsymbol{f}(\boldsymbol{z}^T)]$ and similarly for other functions.

**Lemma 3** (Denoising $\boldsymbol{x}$). *Assume we have two models $p(\boldsymbol{x}^{\leq T}, \boldsymbol{z}^{\leq T})$ and $\hat{p}(\boldsymbol{x}^{\leq T}, \hat{\boldsymbol{z}}^{\leq T})$ as specified in Section 3.1 with parameters $(\boldsymbol{g}, \boldsymbol{f}, G, \sigma^2)$ and $(\hat{\boldsymbol{g}}, \hat{\boldsymbol{f}}, \hat{G}, \hat{\sigma}^2)$, respectively. Assume $d_z < d_x$. Therefore, whenever $\int p(\boldsymbol{x}^{\leq T}, \boldsymbol{z}^{\leq T}) d\boldsymbol{z}^{\leq T} = \int \hat{p}(\boldsymbol{x}^{\leq T}, \hat{\boldsymbol{z}}^{\leq T}) d\hat{\boldsymbol{z}}^{\leq T}$ for all $\boldsymbol{x}^{\leq T}$, we have that the distributions of $\boldsymbol{y}^{\leq T} := \boldsymbol{f}(\boldsymbol{z}^{\leq T})$ and $\hat{\boldsymbol{y}}^{\leq T} := \hat{\boldsymbol{f}}(\hat{\boldsymbol{z}}^{\leq T})$ are equal.*

*Proof.* Let $\mathbb{P}_{\boldsymbol{x}^{\leq T}}$ and $\mathbb{P}_{\hat{\boldsymbol{x}}^{\leq T}}$ be the probability measures respectively induced by the densities $p(\boldsymbol{x}^{\leq T}) := \int p(\boldsymbol{x}^{\leq T}, \boldsymbol{z}^{\leq T}) d\boldsymbol{z}^{\leq T}$ and $\hat{p}(\hat{\boldsymbol{x}}^{\leq T}) := \int \hat{p}(\hat{\boldsymbol{x}}^{\leq T}, \hat{\boldsymbol{z}}^{\leq T}) d\hat{\boldsymbol{z}}^{\leq T}$. We can thus write

$$\mathbb{P}_{\boldsymbol{x}^{\leq T}} = \mathbb{P}_{\hat{\boldsymbol{x}}^{\leq T}}. \tag{17}$$

Let us define $\boldsymbol{y}^t := \boldsymbol{f}(\boldsymbol{z}^t)$ and $\hat{\boldsymbol{y}}^t := \hat{\boldsymbol{f}}(\hat{z}^t)$ for all $t$ where $\boldsymbol{z}^{\leq T} \sim p(\boldsymbol{z}^{\leq T})$ and $\hat{\boldsymbol{z}}^{\leq T} \sim \hat{p}(\hat{\boldsymbol{z}}^{\leq T})$. Let $\mathbb{P}_{\boldsymbol{y}^{\leq T}}$ and $\mathbb{P}_{\hat{\boldsymbol{y}}^{\leq T}}$ be the probability distributions of $\boldsymbol{y}^{\leq T}$ and $\hat{\boldsymbol{y}}^{\leq T}$, respectively. Notice how we can write $\boldsymbol{x}^{\leq T} = \boldsymbol{y}^{\leq T} + \boldsymbol{n}^{\leq T}$ and $\hat{\boldsymbol{x}}^{\leq T} = \boldsymbol{y}^{\leq T} + \hat{\boldsymbol{n}}^{\leq T}$, where $\boldsymbol{n}^t \sim \mathcal{N}(0, \sigma^2 I_{d_x})$ and $\hat{\boldsymbol{n}}^t \sim \mathcal{N}(0, \hat{\sigma}^2 I_{d_x})$ for all $t \leq T$, where the noises are mutually independent across time. This means we can write

$$\mathbb{P}_{\boldsymbol{y}^{\leq T}} * \mathbb{P}_{\boldsymbol{n}^{\leq T}} = \mathbb{P}_{\hat{\boldsymbol{y}}^{\leq T}} * \mathbb{P}_{\hat{\boldsymbol{n}}^{\leq T}}, \tag{18}$$

where $\mathbb{P}_{\boldsymbol{n}^{\leq T}}$ stands for the probability measure of the Gaussian noise (similarly for $\mathbb{P}_{\hat{\boldsymbol{n}}^{\leq T}}$) and $*$ stands for the convolution operator on measures.

The following step makes use of the Fourier transform $\mathcal{F}$ generalized to arbitrary probability measures. See Pollard (67, Chapter 8). Note that the Fourier transform of a probability distribution is exactly the *characteristic function* of the random variable it represents.

$$\mathcal{F}(\mathbb{P}_{\boldsymbol{y}^{\leq T}} * \mathbb{P}_{\boldsymbol{n}^{\leq T}}) = \mathcal{F}(\mathbb{P}_{\hat{\boldsymbol{y}}^{\leq T}} * \mathbb{P}_{\hat{\boldsymbol{n}}^{\leq T}}) \tag{19}$$

$$\mathcal{F}(\mathbb{P}_{\boldsymbol{y}^{\leq T}})\mathcal{F}(\mathbb{P}_{\boldsymbol{n}^{\leq T}}) = \mathcal{F}(\mathbb{P}_{\hat{\boldsymbol{y}}^{\leq T}})\mathcal{F}(\mathbb{P}_{\hat{\boldsymbol{n}}^{\leq T}}) \tag{20}$$

$$\mathcal{F}(\mathbb{P}_{\boldsymbol{y}^{\leq T}})(\omega)e^{-\frac{\sigma^2}{2}\omega^\top\omega} = \mathcal{F}(\mathbb{P}_{\hat{\boldsymbol{y}}^{\leq T}})(\omega)e^{-\frac{\hat{\sigma}^2}{2}\omega^\top\omega}, \forall\omega \in \mathbb{R}^{T \cdot d_x}, \tag{21}$$

where we used the fact that (i) the Fourier transform of a convolution is the product of the Fourier transforms and (ii) the fact that the Fourier transform of a Gaussian random vector with mean 0 and covariance $\sigma^2 I$ is $e^{-\frac{\sigma^2}{2}\omega^\top\omega}$.

Now our goal is to show that $\sigma^2 = \hat{\sigma}^2$. Assume this is false, i.e. $\sigma^2 < \hat{\sigma}^2$ (w.l.o.g.). Because this Fourier transform is positive for all $\omega \in \mathbb{R}^{T \cdot d_x}$, we can divide by its value on both sides and obtain

$$\mathcal{F}(\mathbb{P}_{\boldsymbol{y}^{\leq T}})(\omega) = \mathcal{F}(\mathbb{P}_{\hat{\boldsymbol{y}}^{\leq T}})(\omega)e^{-\frac{\hat{\sigma}^2-\sigma^2}{2}\omega^\top\omega}, \forall\omega \in \mathbb{R}^{T \cdot d_x}, \tag{22}$$

where we recognize that $e^{-\frac{\hat{\sigma}^2-\sigma^2}{2}\omega^\top\omega}$ is the Fourier transform of a Gaussian distribution with mean zero and covariance $(\hat{\sigma}^2 - \sigma^2)I_{T \cdot d_x}$. Now notice how the l.h.s. is the Fourier transform of a distribution with support contained in $\boldsymbol{f}(\mathbb{R}^{Td_z})$, which is a $d_z$-dimensional manifold embedded in $\mathbb{R}^{Td_x}$. Since we assume $d_z < d_x$, the set $\boldsymbol{f}(\mathbb{R}^{Td_z})$ is a proper subset of $\mathbb{R}^{Td_x}$ (i.e. $\boldsymbol{f}(\mathbb{R}^{Td_z}) \neq \mathbb{R}^{Td_x}$). In contrast, the r.h.s. is the Fourier transform of a distribution with full support, i.e. $\mathbb{R}^{Td_x}$, since it is the convolution of $\mathbb{P}_{\hat{y}^{\leq T}}$ and a Gaussian distribution. This is a contradiction since both supports should be equal. Hence we must have that $\sigma^2 = \hat{\sigma}^2$. We can thus write

$$\mathcal{F}(\mathbb{P}_{\boldsymbol{y}^{\leq T}}) = \mathcal{F}(\mathbb{P}_{\hat{\boldsymbol{y}}^{\leq T}}) \tag{23}$$

$$\mathbb{P}_{\boldsymbol{y}^{\leq T}} = \mathbb{P}_{\hat{\boldsymbol{y}}^{\leq T}}, \tag{24}$$

which concludes the proof. $\qquad\square$

**Proposition 1** (Identifiability of $\boldsymbol{f}$ and $p(\boldsymbol{z}^{\leq T})$ up to diffeomorphism). *Assume we have two models $p(\boldsymbol{x}^{\leq T}, \boldsymbol{z}^{\leq T})$ and $\hat{p}(\boldsymbol{x}^{\leq T}, \hat{\boldsymbol{z}}^{\leq T})$ as specified in Section 3.1 with parameters $(\boldsymbol{g}, \boldsymbol{f}, G, \sigma^2)$ and $(\hat{\boldsymbol{g}}, \hat{\boldsymbol{f}}, \hat{G}, \hat{\sigma}^2)$, respectively. Assume further that $\boldsymbol{f}$ and $\hat{\boldsymbol{f}}$ are diffeomorphisms onto their respective images and that $d_z < d_x$ (we do not assume single-parent decoding). Therefore, whenever $\int p(\boldsymbol{x}^{\leq T}, \boldsymbol{z}^{\leq T})d\boldsymbol{z}^{\leq T} = \int \hat{p}(\boldsymbol{x}^{\leq T}, \hat{\boldsymbol{z}}^{\leq T})d\hat{\boldsymbol{z}}^{\leq T}$ for all $\boldsymbol{x}^{\leq T}$, we have $\boldsymbol{f}(\mathbb{R}^{d_z}) = \hat{\boldsymbol{f}}(\mathbb{R}^{d_z})$ and $\boldsymbol{v} := \boldsymbol{f}^{-1} \circ \hat{\boldsymbol{f}}$ is a diffeomorphism. Moreover, the density of the ground-truth latents $p(\boldsymbol{z}^{\leq T})$ and the density of the learned latents $\hat{p}(\hat{\boldsymbol{z}}^{\leq T})$ are related via*

$$p(\boldsymbol{v}(\hat{\boldsymbol{z}}^{\leq T}))\prod_{t=1}^{T}|\det D\boldsymbol{v}(\hat{\boldsymbol{z}}^t)| = \hat{p}(\hat{\boldsymbol{z}}^{\leq T}), \forall\hat{\boldsymbol{z}}^{\leq T} \in \mathbb{R}^{d_z \times T}.$$

*Proof.* By Lemma 3, we have that

$$\mathbb{P}_{\boldsymbol{y}^{\leq T}} = \mathbb{P}_{\hat{\boldsymbol{y}}^{\leq T}}. \tag{25}$$

By marginalizing out $\boldsymbol{y}^{2:T}$ on both sides, we get

$$\mathbb{P}_{\boldsymbol{y}^1} = \mathbb{P}_{\hat{\boldsymbol{y}}^1}, \tag{26}$$

which of course implies that their supports are equal:

$$\text{supp}(\mathbb{P}_{\boldsymbol{y}^1}) = \text{supp}(\mathbb{P}_{\hat{\boldsymbol{y}}^1}) \tag{27}$$

$$\boldsymbol{f}(\text{supp}(p(\boldsymbol{z}^t))) = \hat{\boldsymbol{f}}(\text{supp}(\hat{p}(\hat{\boldsymbol{z}}^t))) \tag{28}$$

$$\boldsymbol{f}(\mathbb{R}^{d_z}) = \hat{\boldsymbol{f}}(\mathbb{R}^{d_z}), \tag{29}$$

where $\text{supp}(\cdot)$ stands for support of a distribution. Note that the last step is because $p(z^1)$ is assumed to have full support (Section Section 3.1).

Equation (29) and the fact that both $\boldsymbol{f}$ and $\hat{\boldsymbol{f}}$ are diffeomorphisms onto their image implies that the map $\boldsymbol{v} := \boldsymbol{f}^{-1} \circ \hat{\boldsymbol{f}}$ is both well-defined and a diffeomorphism.

From (25), we get

$$\mathbb{P}_{\boldsymbol{z}^{\leq T}} \circ \boldsymbol{f}^{-1} = \mathbb{P}_{\hat{\boldsymbol{z}}^{\leq T}} \circ \hat{\boldsymbol{f}} \tag{30}$$

$$\mathbb{P}_{\boldsymbol{z}^{\leq T}} \circ \boldsymbol{f}^{-1} \circ \hat{\boldsymbol{f}} = \mathbb{P}_{\hat{\boldsymbol{z}}^{\leq T}} \tag{31}$$

$$\mathbb{P}_{\boldsymbol{z}^{\leq T}} \circ \boldsymbol{v} = \mathbb{P}_{\hat{\boldsymbol{z}}^{\leq T}} . \tag{32}$$

The above equation combined with the change-of-variable formula yields

$$p(\boldsymbol{v}(\hat{\boldsymbol{z}}^{\leq T})) \prod_{t=1}^{T} |\det D\boldsymbol{v}(\hat{\boldsymbol{z}}^t)| = \hat{p}(\hat{\boldsymbol{z}}^{\leq T}), \ \forall \hat{\boldsymbol{z}}^{\leq T} \in \mathbb{R}^{d_z \times T} , \tag{33}$$

which concludes the proof. $\qquad \square$

**Proposition 2** (Identifying latents of $\boldsymbol{f}$). *Let* $\boldsymbol{f} : \mathbb{R}^{d_z} \to \mathbb{R}^{d_x}$ *and* $\hat{\boldsymbol{f}} : \mathbb{R}^{d_z} \to \mathbb{R}^{d_x}$ *be two diffeomorphisms onto their image* $\boldsymbol{f}(\mathbb{R}^{d_z}) = \hat{\boldsymbol{f}}(\mathbb{R}^{d_z})$. *Assume both* $\boldsymbol{f}$ *and* $\hat{\boldsymbol{f}}$ *have a single-parent decoding structure, i.e.* $|pa_j^F| \leq 1$ *and* $|pa_j^{\hat{F}}| \leq 1$. *Then, the map* $\boldsymbol{v} := \boldsymbol{f}^{-1} \circ \hat{\boldsymbol{f}}$ *has the following property: there exists a permutation* $\pi$ *such that, for all* $i$, *the function* $\boldsymbol{v}_i(\boldsymbol{z})$ *depends only on* $\boldsymbol{z}_{\pi(i)}$.

*Proof.* We see that $\boldsymbol{v} = \boldsymbol{f}^{-1} \circ \hat{\boldsymbol{f}}$ implies

$$\hat{\boldsymbol{f}} = \boldsymbol{f} \circ \boldsymbol{v} . \tag{34}$$

Taking the derivative on both sides of Equation 34, write

$$D\hat{\boldsymbol{f}}(\boldsymbol{z}) = D(\boldsymbol{f} \circ \boldsymbol{v})(\boldsymbol{z}) = D\boldsymbol{f}(\boldsymbol{v}(\boldsymbol{z}))D\boldsymbol{v}(\boldsymbol{z}), \tag{35}$$

where the second equality follows from applying the chain rule, each Jacobian $D\hat{\boldsymbol{f}}(\boldsymbol{z})$ and $D\boldsymbol{f}(\boldsymbol{v}(\boldsymbol{z}))$ is a $d_x \times d_z$ matrix and the Jacobian $D\boldsymbol{v}(\boldsymbol{z})$ is a $d_z \times d_z$ matrix.

In words, Equation 34 tells us that the mapping $\hat{\boldsymbol{f}}$ is "imitating" the mapping $\boldsymbol{f}$ in the following sense: Evaluating $\hat{\boldsymbol{f}}$ is the same as first evaluating $\boldsymbol{v}$ and then evaluating $\boldsymbol{f}$.

We need to show that $\boldsymbol{v}(\boldsymbol{z})$ is a permutation-scaling transformation in the sense defined in the statement of this proposition. To achieve this, we will show that the Jacobian $D\boldsymbol{v}(\boldsymbol{z})$ is a permutation-scaling matrix for all $\boldsymbol{z}$.

We show this in a number of steps:

**Step 1.** Since $\hat{\boldsymbol{f}}$ is a diffeomorphism, its Jacobian, $D\hat{\boldsymbol{f}}$, has full column rank everywhere. Thus, its Moore-Penrose inverse (also known as its pseudo-inverse), $D\hat{\boldsymbol{f}}(\boldsymbol{z})^+$, can be written as,

$$D\hat{\boldsymbol{f}}(\boldsymbol{z})^+ = (D\hat{\boldsymbol{f}}(\boldsymbol{z})^\top D\hat{\boldsymbol{f}}(\boldsymbol{z}))^{-1}D\hat{\boldsymbol{f}}(\boldsymbol{z})^\top. \tag{36}$$

Further, $D\hat{\boldsymbol{f}}(\boldsymbol{z})^+$ is a left inverse; that is, $D\hat{\boldsymbol{f}}(\boldsymbol{z})^+ D\hat{\boldsymbol{f}}(\boldsymbol{z}) = I$.

We can left-multiply both sides of Equation 35 by $D\hat{\boldsymbol{f}}(\boldsymbol{z})^+$, yielding,

$$I = (D\hat{\boldsymbol{f}}(\boldsymbol{z})^\top D\hat{\boldsymbol{f}}(\boldsymbol{z}))^{-1}D\hat{\boldsymbol{f}}(\boldsymbol{z})^\top D\boldsymbol{f}(\boldsymbol{v}(\boldsymbol{z}))D\boldsymbol{v}(\boldsymbol{z}). \tag{37}$$

**Step 2.** We now show that the matrix $D\hat{\boldsymbol{f}}(\boldsymbol{z})^\top D\hat{\boldsymbol{f}}(\boldsymbol{z})$ is diagonal. To see this, consider $k \neq k'$ and write

$$(D\hat{\boldsymbol{f}}(\boldsymbol{z})^\top D\hat{\boldsymbol{f}}(\boldsymbol{z}))_{k,k'} = \sum_{d=1}^{d_x} D\hat{\boldsymbol{f}}(\boldsymbol{z})_{d,k} D\hat{\boldsymbol{f}}(\boldsymbol{z})_{d,k'}. \tag{38}$$

This must be zero since, otherwise, it would imply that there exists a $d$ such that both $D\hat{\boldsymbol{f}}(\boldsymbol{z})_{d,k}$ and $D\hat{\boldsymbol{f}}(\boldsymbol{z})_{d,k'}$ are different from zero, but this is impossible since $y_d$ has only one parent in the graph $F$ (by the "single-parent property").

Define $\Lambda(\boldsymbol{z}) := D\hat{\boldsymbol{f}}(\boldsymbol{z})^\top D\hat{\boldsymbol{f}}(\boldsymbol{z})$, which we just showed is diagonal. Equation 37 implies that

$$D\boldsymbol{v}(\boldsymbol{z})^{-1} = \Lambda(\boldsymbol{z})^{-1}D\hat{\boldsymbol{f}}(\boldsymbol{z})^\top D\boldsymbol{f}(\boldsymbol{v}(\boldsymbol{z})), \tag{39}$$

and since $D\boldsymbol{v}(\boldsymbol{z})^{-1}$ is invertible, $D\hat{\boldsymbol{f}}(\boldsymbol{z})^\top D\boldsymbol{f}(\boldsymbol{v}(\boldsymbol{z}))$ must also be invertible.

**Key idea.** To show that $D\boldsymbol{v}(\boldsymbol{z})$ is a permutation-scaling matrix, we will show that its inverse is a permutation-scaling matrix. We have already shown that $\Lambda(\boldsymbol{z})$ is a diagonal matrix. Thus, what remains is to show that $D\hat{\boldsymbol{f}}(\boldsymbol{z})^\top D\boldsymbol{f}(\boldsymbol{v}(\boldsymbol{z}))$ is a permutation-scaling matrix.

**Step 3.** For any $d_z \times d_z$ invertible matrix $L$,

$$\det(L) = \sum_{\pi \in \Pi_{d_z}} \text{sign}(\pi) \prod_{k=1}^{d_z} L_{\pi(k),k} \neq 0,$$

where $\Pi_{d_z}$ is the set of $(d_z)$-permutations. This implies that there exists a permutation $\pi \in \Pi_{d_z}$ so that for all $k \leq d_z$, $L_{\pi(k),k} \neq 0$.

Applying this result to the invertible matrix $D\hat{\boldsymbol{f}}(\boldsymbol{z})^\top D\boldsymbol{f}(\boldsymbol{v}(\boldsymbol{z}))$,

$$\exists \pi \in \Pi_{d_z} : \forall k \leq d_z, (D\hat{\boldsymbol{f}}(\boldsymbol{z})^\top D\boldsymbol{f}(\boldsymbol{v}(\boldsymbol{z})))_{\pi(k),k} \neq 0$$

$$\implies \forall k \leq d_z, \sum_{d=1}^{d_x} D\hat{\boldsymbol{f}}(\boldsymbol{z})_{d,\pi(k)} D\boldsymbol{f}(\boldsymbol{v}(\boldsymbol{z}))_{d,k} \neq 0 \tag{40}$$

$$\implies \forall k \leq d_z, \exists d_1 : D\hat{\boldsymbol{f}}(\boldsymbol{z})_{d_1,\pi(k)} \neq 0 \neq D\boldsymbol{f}(\boldsymbol{v}(\boldsymbol{z}))_{d_1,k}.$$

**Key idea.** To show that $D\hat{\boldsymbol{f}}(\boldsymbol{z})^\top D\boldsymbol{f}(\boldsymbol{v}(\boldsymbol{z}))$ is a permutation-scaling matrix, we want to show that this matrix has nonzero values *only* at entries of the form $(\pi(k), k)$. We will prove this by contradiction, using the existence of the observed feature $y_{d_1}$ from Equation 40.

**Step 4.** By contradiction, suppose there exists a pair of indices $(k, k')$ such that $k' \neq \pi(k)$ and

$$(D\hat{\boldsymbol{f}}(\boldsymbol{z})^\top D\boldsymbol{f}(v(\boldsymbol{z})))_{k',k} = \sum_{d=1}^{d_x} D\hat{\boldsymbol{f}}(\boldsymbol{z})_{k',d}^\top D\boldsymbol{f}(\boldsymbol{v}(\boldsymbol{z}))_{d,k} \neq 0. \tag{41}$$

This means that there exists a feature $y_{d_2}$ so that

$$D\hat{\boldsymbol{f}}(\boldsymbol{z})_{d_2,k'} \neq 0 \neq D\boldsymbol{f}(\boldsymbol{v}(\boldsymbol{z}))_{d_2,k}. \tag{42}$$

We know that $d_2$ is not equal to $d_1$ from Equation 40, since if $d_1$ and $d_2$ were the same index, then $y_{d_1}$ would have two distinct parents if $\hat{\boldsymbol{f}}$, namely $\pi(k)$ and $k'$.

By selecting only columns $d_1$ and $d_2$ in Equation 35, we obtain the following equality

$$D\hat{\boldsymbol{f}}(\boldsymbol{z})_{\{d_1,d_2\},\cdot} = D\boldsymbol{f}(\boldsymbol{v}(\boldsymbol{z}))_{\{d_1,d_2\},\cdot} D\boldsymbol{v}(\boldsymbol{z}). \tag{43}$$

By (40) & (42) and the fact that both $\boldsymbol{f}$ and $\hat{\boldsymbol{f}}$ satisfy the "single-parent property", we have

$$D\hat{\boldsymbol{f}}(\boldsymbol{z})_{\{d_1,d_2\},\cdot} = \begin{matrix} d_1 \\ d_2 \end{matrix} \begin{pmatrix} 0 & \cdots & 0 & \overset{\pi(k)}{*} & \overset{k'}{0} & 0 & \cdots & 0 \\ 0 & \cdots & 0 & 0 & * & 0 & \cdots & 0 \end{pmatrix} \tag{44}$$

$$D\boldsymbol{f}(\boldsymbol{v}(\boldsymbol{z}))_{\{d_1,d_2\},\cdot} = \begin{matrix} d_1 \\ d_2 \end{matrix} \begin{pmatrix} 0 & \cdots & 0 & \overset{k}{*} & 0 & \cdots & 0 \\ 0 & \cdots & 0 & * & 0 & \cdots & 0 \end{pmatrix}, \tag{45}$$

where $*$ denotes nonzero entries. From the above, it is clear that $D\hat{\boldsymbol{f}}(\boldsymbol{z})_{\{d_1,d_2\},\cdot}$ has a rank of 2 and $D\boldsymbol{f}(\boldsymbol{v}(\boldsymbol{z}))_{\{d_1,d_2\},\cdot}$ has a rank of 1. Since $D\boldsymbol{v}(\boldsymbol{z})$ is invertible we have that the r.h.s. of (43) has rank 1. Equation (43) is thus a contradiction since the l.h.s. has rank 2.

Thus, $(D\hat{\boldsymbol{f}}(\boldsymbol{z})^\top D\boldsymbol{f}(\boldsymbol{v}(\boldsymbol{z})))_{k',k} \neq 0$ if and only if $k' = \pi(k)$, where $\pi$ is a $(d_z)$-permutation. In other words, $D\hat{\boldsymbol{f}}(\boldsymbol{z})^\top D\boldsymbol{f}(\boldsymbol{v}(\boldsymbol{z}))$ is a permutation-scaling matrix.

Going back to Equation 39, $D\boldsymbol{v}(\boldsymbol{z})^{-1}$ is a permutation-scaling matrix since it is a product of a diagonal matrix and a permutation-scaling matrix. Thus, its inverse, $D\boldsymbol{v}(\boldsymbol{z})$ is also a permutation-scaling matrix.

The argument above holds *point-wise*, i.e. for each $\boldsymbol{z}$. However, *a priori*, it is possible that the permutation $\pi$ changes for different values of $\boldsymbol{z}$. It turns out this is impossible since that would violate the fact that $D\boldsymbol{v}(\boldsymbol{z})$ is continuous. $\quad\square$

## C  Optimization

### C.1  Augmented Lagrangian Method

The main idea of the Augmented Lagrangian Method (ALM) is to remove constraints in an optimization problem and instead add a penalty term to the objective (For more detailed explanations, see Nocedal and Wright (60), Chapter 17). For an optimization problem where we aim to minimize $\mathcal{L}(x)$ subject to a constraint $h(\boldsymbol{x}) = 0$, the equivalent ALM objective is:

$$L(\boldsymbol{x}, \lambda_k, \mu_k) = \mathcal{L}(\boldsymbol{x}) + \lambda_k h(\boldsymbol{x}) + \frac{\mu_k}{2}||h(\boldsymbol{x})||^2.$$

To approximately solve the constrained problem, a sequence of problems will be solved where $\lambda_k$ and $\mu_k$ will be incrementally increased as $k$ increases. A problem is considered solved when the loss on a held-out dataset does not decrease. Then, a new problem is initialized with the values $\lambda_{k+1}$ and $\mu_{k+1}$:

$$\lambda_{k+1} \leftarrow \lambda_k + \mu_k \cdot h(W^*_{(k)})$$

$$\mu_{k+1} \leftarrow \begin{cases} \eta \cdot \mu_k, & \text{if } h(W^*_{(k)}) > \delta \cdot h(W^*_{(k-1)}) \\ \mu_k, & \text{otherwise} \end{cases}$$

where $\mu_k > 0$ and $\eta > 1$ and $W^*_{(k)}$ is the approximate solution to the $k$-th problem.

### C.2  Projected Gradient

The projected gradient descent method can be used when some parameters are under constraints. In general, we proceed in two steps: 1) perform a step of gradient descent and 2) project the result on the feasible set. More formally, we do a normal gradient descent step from $x_k$ :

$$y_{k+1} = x_k - \alpha \nabla f(x_k),$$

and then we project the result $y_{k+1}$ on the feasible set $\mathcal{Q}$:

$$x_{k+1} = \arg\min_{x \in \mathcal{Q}} \frac{1}{2}||x - y_{k+1}||_2^2.$$

In our case, the feasible set is $\mathbb{R}_{\geq 0}$, thus the projection is simply:

$$x_{k+1} = \begin{cases} y_{k+1} & \text{if } y_{k+1} \geq 0 \\ 0 & \text{otherwise.} \end{cases} \tag{46}$$

## D  Datasets

### D.1  Synthetic Datasets

Here, we detail the generative process of the synthetic datasets. The four main steps are:

1. Sample transition graphs $G \in \{0,1\}^{\tau \times d_z \times d_z}$.
2. Sample mechanisms to generate the latent variables $\boldsymbol{z}$ with relations following $G$.
3. Sample a matrix $W$.
4. Sample mechanisms of the decoding function to generate the observable $\boldsymbol{x}$ from $\boldsymbol{z}$.

**1- Graphs G**. For the adjacency matrices representing the lagged relations, we sampled independently $G^k_{ij} \sim Bernoulli(p)$ where $p \in [0,1]$ is a parameter corresponding to the probability of adding an edge. As a reminder: $G^k_{ij} = 1$ if and only if $z^{t-k}_j$ is a parent of $z^t_i$. Note that since these adjacency matrices represent links at different timesteps, we don't need to sample DAGs since it is impossible to create cycles.

For the graph $G^1$, we systematically set the elements of its diagonal to 1. These edges correspond to the relations of $z^t_i$ to themselves at the previous timestep: $z^{t-1}_i$. This assumption is often observed in real-world phenomena and is commonly used to generate time-series datasets (78; 49).

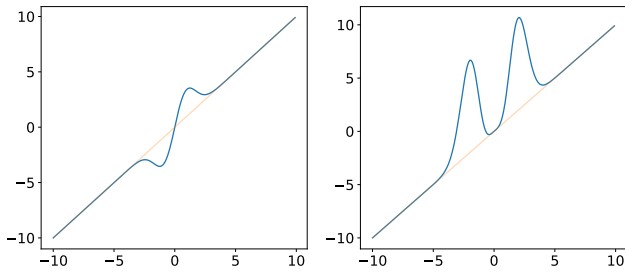

Figure 6: Examples of nonlinear functions used for the dynamics. For large values, the functions behave as linear functions.

**2- Dynamic's functions**. We first explain how we generate the mechanisms for the *nonlinear* case and, since it is a particular case, we then explain the *linear* case.

It is not obvious in general how to sample a nonlinear generative process that is stationary or, at least, that won't greatly diverge over time. In order to have non-divergent generative processes with nonlinear functions, we follow (49) (See Theorem 1). Their method relies on two tricks: 1) the process is additive and the functions used have a linear behavior for large values, and 2) the coefficients are chosen so that an equivalent linear process would be stationary.

We use the following structural equation:

$$z_i^t := \sum_{k=0}^{\tau} \sum_{j=1}^{d_z} G_{ij}^k A_{ij}^k s_{ij}^k(z_j^{t-k}) + \epsilon_i,$$

where $\epsilon_i \sim \mathcal{N}(0, 1)$ and the coefficients $A_{ij}$ are independently sampled from $\mathcal{U}([-1, -0.2] \cup [0.2, 1])$ and will be reweighted to ensure stationary. The range around $0$ is removed to ensure that the data is faithful to the graph. Following (78), each nonlinear dynamic's function $s_{ij}^k$ has a structure similar to:

$$f_1(x) = x(1 + 4e^{\frac{-x^2}{2}})$$

$$f_2(x) = x(1 + 4x^3 e^{\frac{-x^2}{2}}).$$

See the code to see all the functions that were used. As it can be noticed in Fig. 6, these functions are almost linear when $x$ is large. Then, to make the process stationary, we have to make sure that the linear process that has the same coefficient matrix $A$ would be stationary. One way to verify this is to make sure that all the eigenvalues of the following matrix $H$ have a modulus of less than one:

$$H = \begin{bmatrix} A^\tau & A^{\tau-1} & \ldots & A^0 \\ I_\tau & 0 & \ldots & 0 \\ 0 & \ldots & 0 & \vdots \\ 0 & 0 & I_\tau & 0 \end{bmatrix} \tag{47}$$

To do so, we compute the spectrum $\rho(H)$ and we divide each matrix $A^k$ by $\rho(H)^{k+1}$. The resulting process will be stationary (49).

For the *linear* datasets, we follow the same sampling process except that $s_{ij}^k$ is the identity.

**3- Graph F**. To generate the observable $x$ from $z$, we first sample the matrix $W \in \mathbb{R}_{\geq 0}^{d_x \times d_z}$ which is non-negative and orthogonal following these steps:

1. We first sample an assignment matrix $M \in \{0, 1\}^{d_x \times d_z}$ where in each row, only one element is set to 1 and the rest is equal to 0. We also make sure that for each column, there is at least one element equal to 1.

2. We sample independently $A_{ij} \sim \mathcal{U}([0.2, 1])$ and mask it: $\tilde{W} := M \odot A$.

3. Finally, we normalize the column of $W$ to ensure that it is orthogonal. In other words: $W_{:j} := \frac{\tilde{W}_{:j}}{||\tilde{W}_{:j}||_2}$.

**4- Decoding functions.** In the nonlinear case, each function $r_j$ (from Section 3.3 §3) was randomly sampled either as a linear function or as a nonlinear function. The nonlinear function takes the following general form:

$$f(x) = ax(2\sigma(10x) - 1),$$

where $a \sim \mathcal{U}[0.2, 0.7]$. This nonlinear function is approximately an absolute value function but is still differentiable since it is smooth at $x = 0$. In the linear case, each function $r_j$ is the identity. We sample the observable $\boldsymbol{x}$ following:

$$x_j^t \mid \boldsymbol{z}^t \sim \mathcal{N}(r_j(W\boldsymbol{z}^t), \sigma_j^2),$$

where $\sigma_j^2 = 0.1$ and $0.5$ in experiments with linear and nonlinear decoding, respectively.

## D.2    REAL-WORLD DATASETS

Here, we describe how the sea-level pressure data was regridded to an icosahedral-hexagonal grid to achieve an equal-area projection (54; 55). This form of regridding is motivated by the fact that the poles are singularities in the traditional longitude-latitude grid, i.e. that each grid cell contains a different amount of total surface area (for $1° \times 1°$ resolution, the length of a grid cell decreases from $\approx 111$ km at the equator to $0$ km at the poles). This means that more grid cells represent a smaller area the further we move from the equator towards the south or north poles, leading to an overrepresentation of polar regions and an underrepresentation of equatorial regions. One way to address this issue is by projecting the data on a geodesic grid, such as the icosahedral-hexagonal grid (80), as nowadays used in climate models (e.g. ICON, see Pham et al. (66)). By regridding the data we can ensure that northern/southern weather regions are not overrepresented in the clustering and during the causal discovery process.

The sea-level pressure data was projected to the GME icosahedral grid (55), with first-order conservative remapping (32) and the number of intervals $NI = 24$ to match the original resolution of $2.5° \times 2.5°$ at the equator as closely as possible. The original netCDF4 input files had to be converted to GRIB2 files for this purpose since netCDF4 files can only store quadrilateral data, whereas GRIB2 has no such restrictions (15). After making the original data GRIB2 compliant and converting it to this format, Climate Data Operators (CDO) (82), was used to remap the longitude-latitude gridded data to the GME icosahedral-hexagonal grid. Two parameters are relevant during the regridding process: 1) $NI$, the number of intervals, and 2) the remapping function. The number of intervals was chosen in a manner to resemble the resolution of the original data at the equator, and in a manner to allow the recursive generation of bisected equilateral triangles (90). The remapping function was chosen to ensure a monotonic remapping that can deal with a fine-to-coarse setting. First-order or second-order conservative remapping seem to be the best choices for that and we decided to use first-order conservative remapping since it is sufficiently accurate and easier to handle. The regridded sea-level pressure files in GRIB2 format can be easily loaded with the xarray package in Python (24).

## E    METHODS AND IMPLEMENTATION DETAILS

### E.1    VARIMAX-PCMCI

Varimax-PCMCI is a two-step method where 1) a dimensionality reduction is conducted to obtain time series of latent variables and then 2) a causal discovery method is applied on the latent level time series. We follow Tibau et al. (87) by using PCA with a Varimax-rotation as the dimensionality reduction method. This method was observed empirically in Tibau et al. (87) to lead to better results. For more details on PCA-Varimax, see Appendix I. For the causal discovery algorithm we use PCMCI+ (75) which is an update to PCMCI, that also supports instantaneous connections. For the conditional independence tests used by PCMCI+, we use the partial correlation test for linear dynamics and the CMI-knn test (74), which is a test based on a nearest-neighbor estimator of conditional mutual information, for nonlinear dynamics. We use the implementation from `https://github.com/jakobrunge/tigramite` and `https://github.com/xtibau/savar/tree/master/savar`.

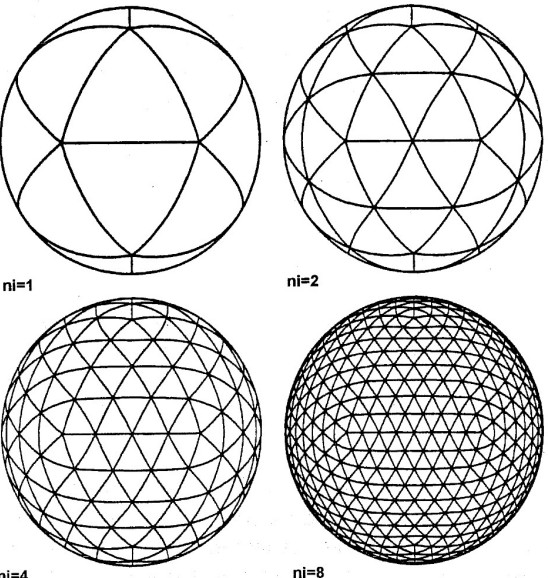

Figure 7: Visualization of icosahedral-hexagonal grids with different $NI$'s (number of intervals on triangle edge). the grid is generated recursively by halving the triangles, resulting in new, finer triangles. The figure stems from Majewski (54).

The significance level in PCMCI+ was set to $\alpha_{\mathrm{PC}} = 0.2$ and the time lags of causal links were restricted to $\tau \geq 1$, hence, only lagged links are considered. PCMCI+ then still slightly differs from PCMCI. Both share the $\mathrm{PC}_1$ phase that detects lagged (supersets) of parents, but they differ in the second phase because the MCI tests are only restricted to the superset of parents found in the first phase in PCMCI+ (75), whereas all lagged links are tested again in case of PCMCI.

## E.2 CDSD

For all neural networks, we use leaky-ReLU as activation functions. For the neural networks $g_j$ fitting the nonlinear dynamic, we used MLPs with 2 hidden layers and 8 hidden units. For the neural network $r_j$ fitting the nonlinear encoding, instead of having $d_x$ functions, we use some parameter sharing in order to keep the number of parameters low. We use a single neural network that receives as input the masked $W z^t$ and an embedding (dimension 10) of the index $s(j)$ is concatenated to the input. This neural network has 2 hidden layers and 32 hidden units. The log of the variance terms and the matrix $W$ are free parameters initialized respectively to $-4$ and $\mathcal{U}[\frac{1}{10d_z}, \frac{1}{d_z}]$. The parameters $\Gamma$ are initialized to 5 which corresponds to an almost full graph (i.e. $\sigma(\Gamma) \approx \mathbf{1}$). We use the optimizer RMSProp (22) with a learning rate of $1e-3$ and batch size of $64$.

## E.3 MCC METRIC

As stated in the main text, the mean coefficient correlation (MCC) is a metric commonly used in causal representation learning to assess the quality of the learned representation. It is necessary since the identifiability result is up to permutation. We use an implementation from Khemakhem et al. (37) (https://github.com/ilkhem/icebeem/blob/master/metrics/mcc.py). This corresponds to calculating the Pearson correlation between the learned representation $\hat{z}$ and the ground-truth $z$ under all possible permutations, selecting the permutation $\pi$ leading to the highest score, and taking its mean. To calculate the SHD between the learned graph $\hat{G}$ and $G$, we first apply the permutation $\pi$ to the learned graph.

# F    HYPERPARAMETER SEARCH

For all experimental conditions, we use the default hyperparameters specified in Table 1 and 2. These values were determined based on the SHD from a few experiments on distinct synthetic datasets than those used for evaluation. The regularisation coefficient (for the graph sparsity of CDSD) and the alpha term (the significance threshold for the conditional independence tests of Varimax-PCMCI) have a bigger impact in terms of performance and accordingly they vary per dataset. For both methods, they have been tested respectively on the log scale $[-3, 1]$ and $[-11, -1]$. For these values, for each experimental condition, 10 datasets (distinct from evaluation) were used each with 10 different values of regularisation and alpha. The average of the parameter value leading to the best SHD of the 10 datasets was used for the synthetic experiments. As stated earlier, both $d_z$ and $\tau$ are given when using synthetic datasets. In the linear dynamics case, Varimax-PCMCI uses the partial correlation test and CDSD uses neural networks $g_j$ without hidden layers. In the nonlinear dynamics case, Varimax-PCMCI uses the CMI-knn test and CDSD uses neural networks $g_j$ without 2 hidden layers and 8 hidden units.

For the real-world dataset, we reused the same default hyperparameters. For the regularisation coefficient, we selected it based on the mean-square error between the prediction $\hat{x}^t$ and $x^t$ on a held-out testing set. The split ratio is 0.8 for the training set and 0.2 for the testing set.

Table 1: Default hyperparameters for CDSD

| CDSD hyperparameters |
| --- |
| **ALM parameters**: 
 threshold: $10^{-4}$, $\mu_0$: $10^{-3}$, $\gamma_0$: 0, $\eta$: 2, $\delta$: 0.9 
 **Optimizer**: 
 RMSProp, learning rate: $10^{-3}$, batch size: 64 
 **Transition NN ($g_j$):** 
 Nonlinearity: Leaky-ReLU, 
 # hidden layers: [linear = 0, nonlinear = 2], 
 # hidden units: 8 
 **Encoder/Decoder NN for nonlinear encoding:** 
 Nonlinearity: Leaky-ReLU, 
 # hidden layers: 2, 
 # hidden units: 32 |

Table 2: Default hyperparameters for PCMCI+

| PCMCI+ hyperparameters |
| --- |
| $\alpha_{PC} = 0.2$ 
 $\tau_{min} = 1$ 
 CI test: linear = partial corr., nonlinear = CMI-knn |

# G    ADDITIONAL EXPERIMENTS

## G.1    RUNNING TIME

We show the running time of CDSD and Varimax-PCMCI as a function of the number of samples and the number of latent variables. Note that the methods have not been highly optimized and that these running times are only given to give a general idea and assess the trends of growth. Varimax-PCMCI is much faster than CDSD when using partial correlation as the conditional independence test (see the left panels of Fig. 8 and 9). However, for the nonlinear dynamics, CDSD is much faster, whereas Varimax-PCMCI often requires more than 24 hours to run. We only included cases where the time was under 24 hours (see the right panels of Fig. 8 and 9).

In Figure 8, it can be observed that for linear dynamics, the running time is almost constant in function of the number of samples. However, for the nonlinear dynamics, the time for Varimax-PCMCI

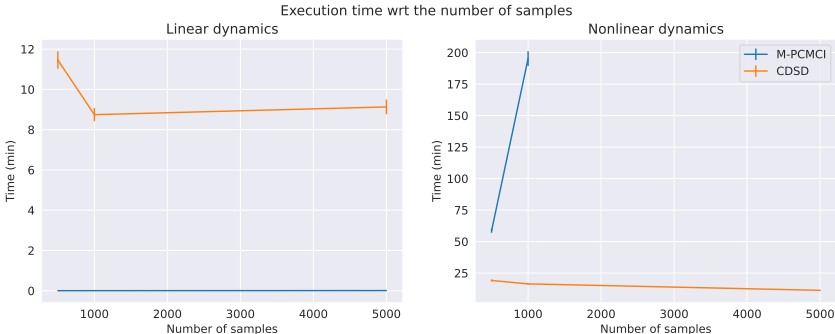

Figure 8: Comparison of the running time with respect to the number of samples in linear and nonlinear dynamics settings.

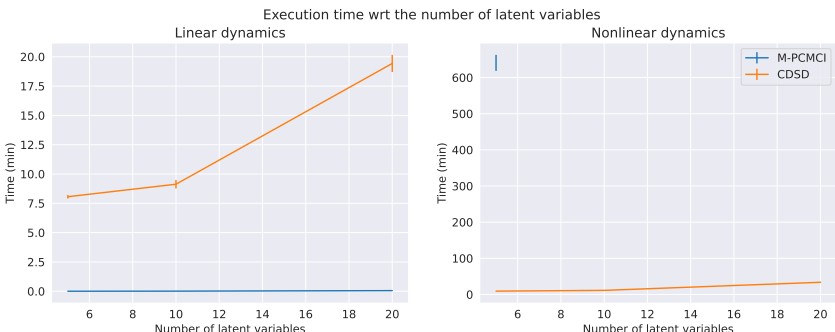

Figure 9: Comparison of the running time with respect to the number of latent variables in linear and nonlinear dynamics settings.

increases steeply and is > 24h for 5000 samples. In Figure 9, it can be observed that, as expected, the running time increases as the number of latent variables of the model increases. The same pattern can be observed for Varimax-PCMCI: while the running time is extremely low for linear functions, it is very high and only below 24 hours when 5 latent variables (with 5000 samples) are considered.

All experiments were run on AMD EPYC 7742 2.25GHz 64-Core Processor with 40G of RAM. We present in Table 3 the total running time for each experiment.

Table 3: Running time for each set of experiments

| Synthetic experiments |
| --- |
| **Linear dynamics, linear decoder:** |
| CDSD: 76 hours |
| Varimax-PCMCI: 0.3 hours |
| **Nonlinear dynamics, linear decoder:** |
| CDSD: 116 hours |
| Varimax-PCMCI: 1212 hours |
| **Linear dynamics, nonlinear decoder:** |
| CDSD: 383 hours |
| Varimax-PCMCI: 0.03 hours |
| **Ablation study:** 4.3 hours |
| **Real-world experiments:** 75 hours |

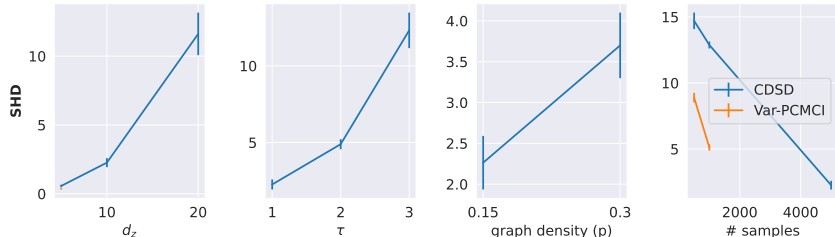

Figure 10: Comparison in terms of SHD (lower is better) on nonlinear dynamics datasets.

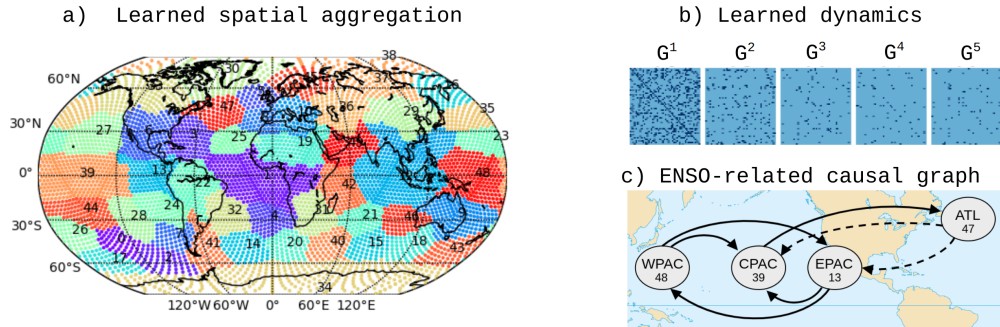

Figure 11: Spatial partitioning learned by CDSD using a lower regularisation. The dashed lines in the causal graph represent edges that were not present in the main text.

### G.2 Synthetic Experiments

We show in Fig. 10 the results of CDSD on all the different experimental conditions in the nonlinear dynamics case. These results were not included in the main text since, as explained in the previous section, Varimax-PCMCI had a running time too high in most conditions. It can be observed that overall the nonlinear dynamics seems to be a more challenging task than its linear counterpart. Otherwise, the general trends are similar (improvement with the number of samples, deterioration as the other parameters increase).

### G.3 Real-world Experiment

In this section, we show additional results on the real-world dataset, namely we show an alternate result using a lower regularisation, we show more clearly the learned clusters and we show additional results using a lower number of latents.

In Fig. 11 we show an alternate result that had a similar validation loss, but learned a denser graph. Overall, the conclusions remain the same. It can be observed that the learned spatial aggregation is nearly identical to the one in the main text. The graph is denser, but the same general pattern is observed: $G^1$ is denser, the diagonal has edges, etc. Furthermore, a similar pattern can be observed between the nodes related to ENSO with two additional edges.

In the main text, we claimed that the learned clusters are localized. Since in the main figure of the real-world application (Fig. 5) some neighboring clusters have very similar colors, for clarity, we show in Fig. 12 all the learned clusters separately. It can be observed that all the clusters are well localized even if this is not directly enforced in CDSD. In contrast, if CDSD without any constraints on W is used, the learned clusters are not localized (Fig. 15 and Fig. 16) and regions specific to ENSO are now all in a unique cluster.

We also investigated using a number of latent relatively smaller ($d_z = 20$). Fifty latents were chosen in the main text since it is close to what has been used in similar studies (60 were used in (78)). In Fig. 13 we present the learned clustering. It can be seen that the coarser aggregation has many regions that are the union of the one observed in Fig. 5, however, interestingly, the regions related to ENSO (WPAC, CPAC,

EPAC, but not ATL) are still separated. Contrary to using fifty latents, some regions are composed of disconnected regions potentially indicating that the number of latent variables is too low, see Fig. 14.

## H  EXTENSIONS OF CDSD

### H.1  USING CDSD IN MULTIVARIATE AND MULTI-SUBJECT SETTINGS

**Multivariate case.** Researchers are often interested in multivariate problems where each feature might require different clustering. For the climate example, researchers might have several features of interest: sea-level pressure, temperature, precipitation, etc. While we only presented the univariate version of CDSD, it can easily be used as a multivariate method by slightly modifying $W$.

Assume we have $d$ features of interest. Reusing the notation from Section 3, let $\boldsymbol{x}_{ki}^t$ and $\boldsymbol{z}_{kj}^t$ be the observable and latent variables at time $t$ pertaining to the $k$-th feature. Note that now the cardinality of $d_x^k$ and $d_z^k$ can vary in function of $k$. As a matter of fact, since they are different features, it might be adequate to model them at a different coarsening. We denote the total dimensionality by $d_x := \sum_{k=1}^d d_x^k$ and $d_z := \sum_{k=1}^d d_z^k$.

For a feature $k$ we assume the observables $\boldsymbol{x}_k$ can only be related to the latent variables $\boldsymbol{z}_k$. This amounts to blacklisting some edges in $W \in \mathbb{R}^{d_z \times d_x}$. Thus, $W$ will be a block matrix of the form:

$$
\begin{bmatrix}
W^1 & & & \\
& W^2 & & \\
& & \ddots & \\
& & & W^k
\end{bmatrix},
\tag{48}
$$

where $W^k \in \mathbb{R}^{d_z^k \times d_x^k}$ is the matrix defining the connections of the variable related to the feature $k$. Besides this change, the orthogonality constraint can be directly applied to each matrix $W^k$.

**Multiple subjects case.** In many scientific applications such as neurosciences, many subjects are observed and it is often assumed that they share common properties. For example, in a brain imaging application, one could assume that the brain regions are invariant, whereas the connectivity between these regions is patient-specific (as in Monti and Hyvärinen (56)). Similarly to the multivariate case, we can use graphs with an extra dimension ($G \in \{0,1\}^{d_p \times \tau \times d_z \times d_z}$ where $d_p$ is the number of patients) and blacklist connections along the dimension related to the patient.

### H.2  SUPPORTING INSTANTANEOUS CAUSAL RELATIONSHIPS

In order to support instantaneous relations, we add two components: 1) a graph $G^0$ and the parameters $\Gamma^0$ to the generative model, and 2) an acyclicity constraint to the objective. Note that contrary to $G^1, \ldots, G^\tau$, the graph $G^0$ has to be a *directed acyclic graph* (DAG) in order to have a valid factorization. We have the following modified transition model:

$$
p(\boldsymbol{z}^t \mid \boldsymbol{z}^{<t}) := \prod_{j=1}^{d_z} p(z_j^t \mid \boldsymbol{z}_{\mathbf{pa}_j^{G^0}}, \boldsymbol{z}^{<t}).
\tag{49}
$$

Each conditional is:

$$
p(z_j^t \mid \boldsymbol{z}_{\mathbf{pa}_j^{G^0}}, \boldsymbol{z}^{<t}) := h(z_j^t; \, g_j([G_{j:}^0 \odot \boldsymbol{z}^t, G_{j:}^1 \odot \boldsymbol{z}^{t-1}, \ldots, G_{j:}^\tau \odot \boldsymbol{z}^{t-\tau}])).
\tag{50}
$$

The observation and the density models remain the same. For the objective, we add the acyclicity constraint (95) on $\Gamma^0$:

$$
\mathrm{Tr}(e^{\sigma(\Gamma^0)}) - d_z = 0.
\tag{51}
$$

This constraint formulation is continuous and thus differentiable. We can then use the quadratic penalty method to optimize this constrained problem (58). However, in this case, the identifiability of the graph $G$ is not guaranteed: at best the Markov Equivalence Class can be recovered or additional assumptions are needed such as assuming an additive noise model.

### H.3 SUPPORTING INTERVENTIONS

In order to support interventions on $z$, the model can be adapted following Brouillard et al. (9); Gao et al. (17); Lei et al. (48). Assume that we have interventional data from $K$ different interventions. The $k$-th interventional distribution of the transition model $p^{(k)}(z^t \mid z^{<t})$ with the intervention target $\mathcal{I}_k \subseteq [d_z]$ is given by:

$$p^{(k)}(z^t \mid z^{<t}) := \prod_{j \in \mathcal{I}_k} p^{(k)}(z_j^t \mid z^{<t}) \prod_{j \notin \mathcal{I}_k} p(z_j^t \mid z^{<t}), \tag{52}$$

where $p^{(k)}$ are conditionals different from the observational conditionals except for $k = 0$ where $\mathcal{I}_0 := \emptyset$. Since there are no interventions on $x$, the conditional $p(x^t \mid z^t)$ remains the same as in the observational case. The joint interventional distribution is:

$$p^{(k)}(x^{\leq T}, z^{\leq T}) := \prod_{t=1}^{T} p^{(k)}(z^t \mid z^{<t}) p(x^t \mid z^t). \tag{53}$$

In terms of ELBO, we now have:

$$\log p^{(k)}(x^{\leq T}) \geq \sum_{t=1}^{T} \Big[ \mathbb{E}_{z^t \sim q(z^t \mid x^t)} \big[ \log p(x^t \mid z^t) \big] - \tag{54}$$

$$\mathbb{E}_{z^{<t} \sim q(z^{<t} \mid x^{<t})} \mathrm{KL} \Big[ q(z^t \mid x^t) \,||\, p^{(k)}(z^t \mid z^{<t}) \Big] \Big]. \tag{55}$$

We denote this ELBO as $\mathcal{L}_x^{(k)}$. Finally, the objective changes to:

$$\max_{W, \Gamma, \phi} \sum_{k=0}^{K} \mathbb{E}_{G \sim \sigma(\Gamma)} \big[ \mathbb{E}_{x \sim p^{(k)}} \big[ \mathcal{L}_x^{(k)}(W, \Gamma, \phi) \big] \big] - \lambda_s ||\sigma(\Gamma)||_1 \tag{56}$$

$$\text{s.t. } W \text{ is orthogonal and non-negative,}$$

where $\phi$ is now augmented with different parameters for each conditional $p^{(k)}$.

## I PCA-VARIMAX

### I.1 PCA

PCA is a commonly used dimensionality reduction method that finds a linear mapping W between the data and a latent projection having a smaller dimensionality. It can be framed as finding the projection that maximizes the variance or, alternatively, as the projection that minimizes the reconstruction loss (64; 23). If we consider the latter formulation, we can write the PCA method as the following optimization problem:

$$L(x; W) = ||x - WW^T x||_2^2$$
$$\hat{W} \in \arg\min_{W \, s.t. \, W^T W = I_{d_z}} L(x; W),$$

where $x \in \mathbb{R}^{d_x}$ and $W \in \mathbb{R}^{d_x \times d_z}$.

In that case, the matrix W is not identifiable since any rotation would lead to the same fit of PCA (as already noted by (3)). To see this, suppose that a rotation matrix $R \in \mathbb{R}^{d_z \times d_z}$ is applied to $W$. The resulting matrix $\tilde{W} = WR$ is still orthogonal (since orthogonal matrices are closed under multiplication) and leads to the same loss since $RR^T = I$:

$$L(x; \tilde{W}) = ||x - \tilde{W}\tilde{W}^T x||_2^2$$
$$= ||x - WRR^T W^T x||_2^2$$
$$= ||x - WW^T x||_2^2.$$

## I.2 THE VARIMAX ROTATION

In practice, researchers from many different fields, such as Earth science (88; 87), use a criterion called Varimax (33) in order to find a particular rotation of W. The optimal rotation according to this criterion is given by:

$$R_{\text{VAR}} = \arg\max_R \Big( \frac{1}{d_z} \sum_{i=1}^{d_x} \sum_{j=1}^{d_z} (WR)_{ij}^4 - \sum_{i=1}^{d_x} \Big( \frac{1}{d_z} \sum_{j=1}^{d_z} (WR)_{ij}^2 \Big)^2 \Big). \tag{57}$$

While it leads to the same loss, this rotation has been shown empirically to lead to more interpretable variables (31; 68) (i.e. the latent dimensions are recognized as important factors of variation by experts in the fields). The criterion corresponds to maximizing the sum of the variances of the squared loadings $W_{ij}^2$. Intuitively, this criterion will make the large loadings larger and the small loadings smaller making the rotated matrix sparse. The good representation learned by PCA-Varimax when $W$ respects the single-parent assumption could be explained by the identifiability results of Zheng et al. (96). They show that, under a more general sparsity assumption than the single-parent assumption, $W$ can be identified using an adequate sparsity regularisation.

## I.3 EMPIRICAL VALIDATION

In this section, we show empirically that PCA-Varimax can learn a disentangled representation when the decoding function is linear and $W$ respects the single-parent assumption. The results stress the fact that PCA without the Varimax rotation generally leads to entangled representation.

The PCA-Varimax algorithm used in Varimax-PCMCI can be decomposed in three steps:

- **Step 1.** Apply PCA and recover a matrix $\hat{W}$.
- **Step 2.** Find the rotation matrix $R_{\text{VAR}}$ that maximizes the Varimax criterion. Apply this matrix to $\hat{W}$:

$$\hat{W}_{\text{VAR}} := \hat{W} R_{\text{VAR}}.$$

- **Step 3.** Apply a reflection matrix $S$:

$$\hat{W}_{\text{VAR}}^+ := \hat{W}_{\text{VAR}} S,$$

where $S_{jj} = \text{sgn}(\arg\max_i |W_{ij}|)$ and $S_{ij} = 0$ for all $i \neq j$.

For this experiment, we do an ablation study of these three steps, namely we consider the three conditions: 1) PCA without rotation (PCA), 2) PCA with the Varimax rotation (PCA-Var), and 3) PCA with the Varimax rotation and reflection (PCA-Var+). We reuse the default synthetic datasets with the same alpha values that were chosen in the main hyperparameter search. For each condition, we present the average on 10 different datasets.

In terms of representation, it can be seen in Figure 17 that the Varimax rotation is essential to learn a disentangled representation as observed by the MCC metric. Learning a good representation has a repercussion on the recovery of the graph $G$ as shown by the SHD metric. The reflection operation might flip signs but does not have any impact in terms of disentanglement. However, if we look at the recovered matrices $\hat{W}$ in terms of absolute error with $W$, the reflection allows us to recover a matrix more similar to $W$.

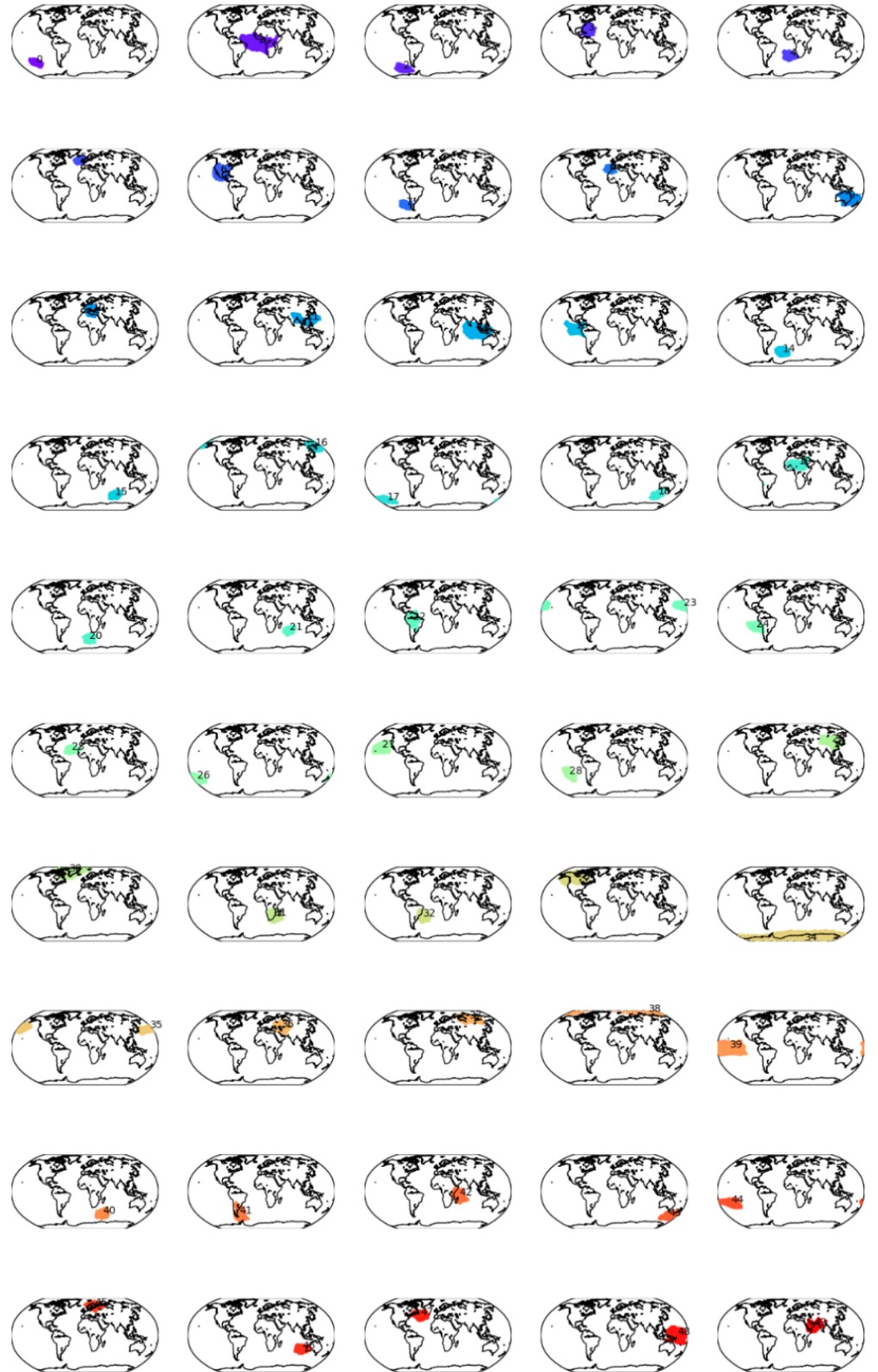

Figure 12: Representation of each region learned by CDSD.

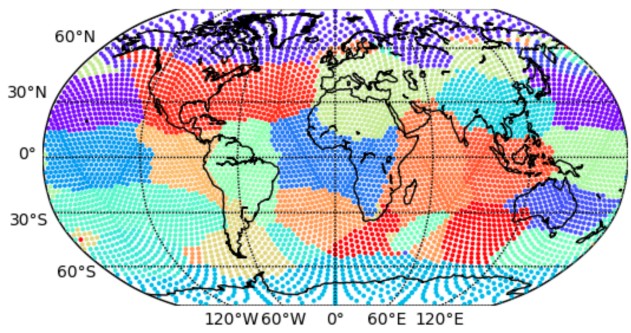

Figure 13: Spatial partitioning learned by CDSD using a low number of latents ($d_z = 20$).

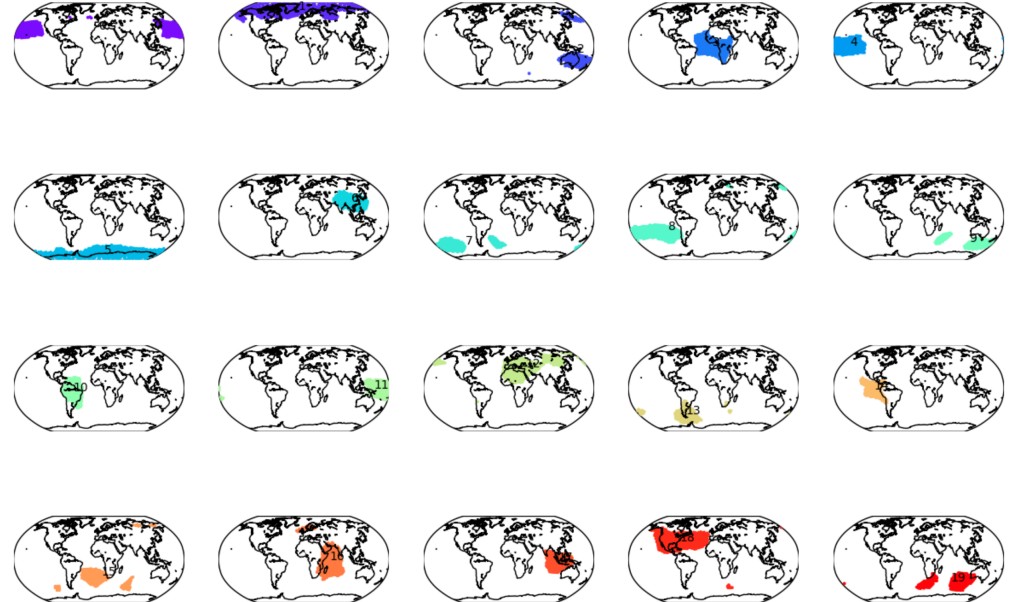

Figure 14: Representation of each region learned by CDSD using a low number of latents ($d_z = 20$).

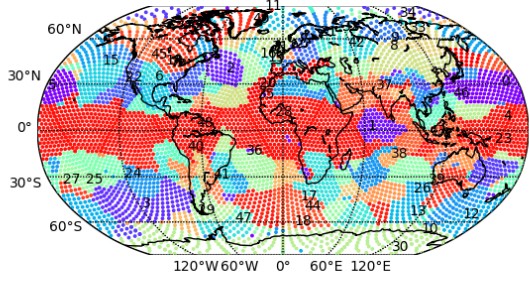

Figure 15: Spatial partitioning learned by CDSD without constraint on $W$ ($d_z = 50$).

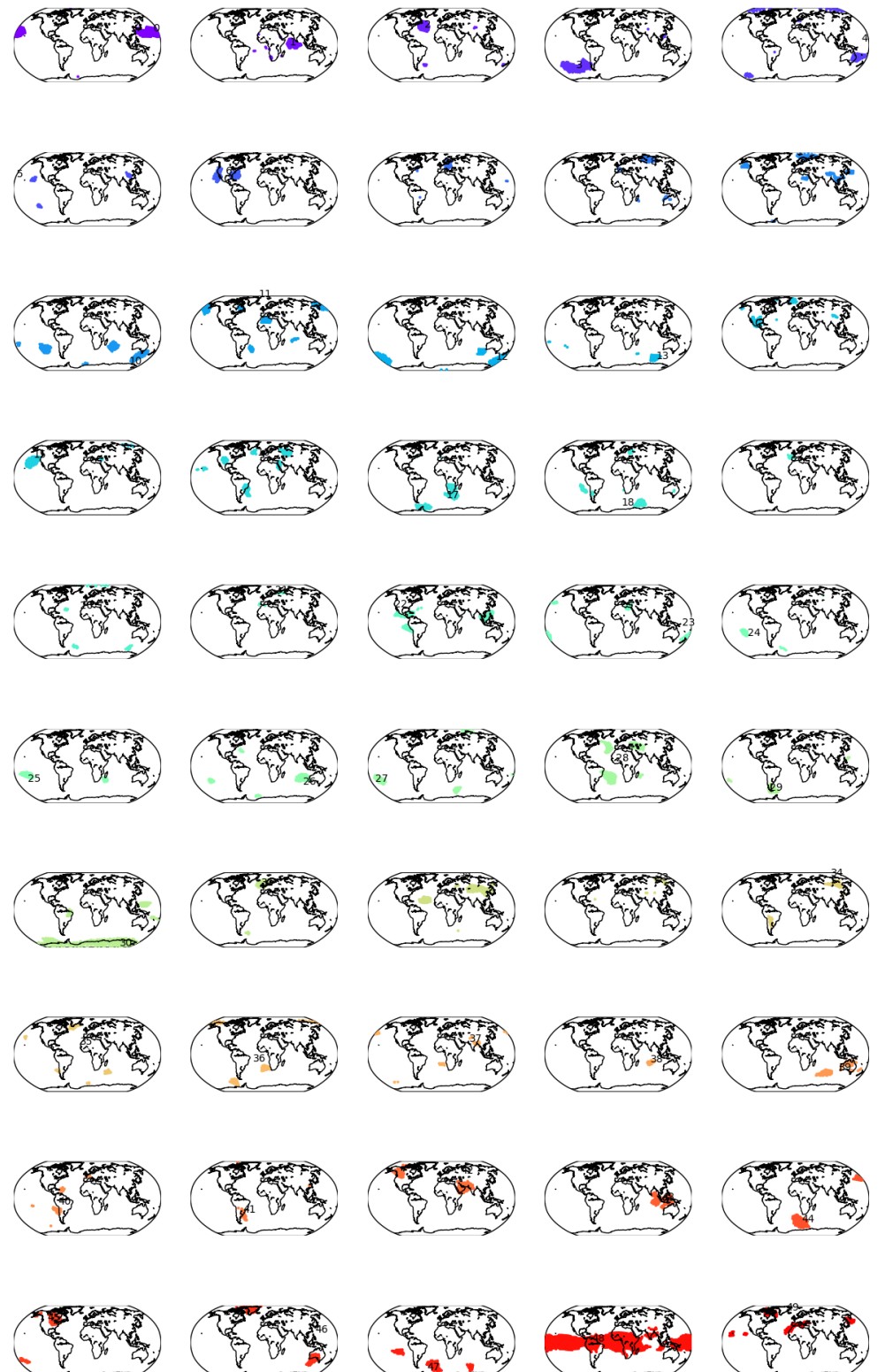

Figure 16: Representation of each region learned by CDSD without constraint on $W$ ($d_z = 50$).

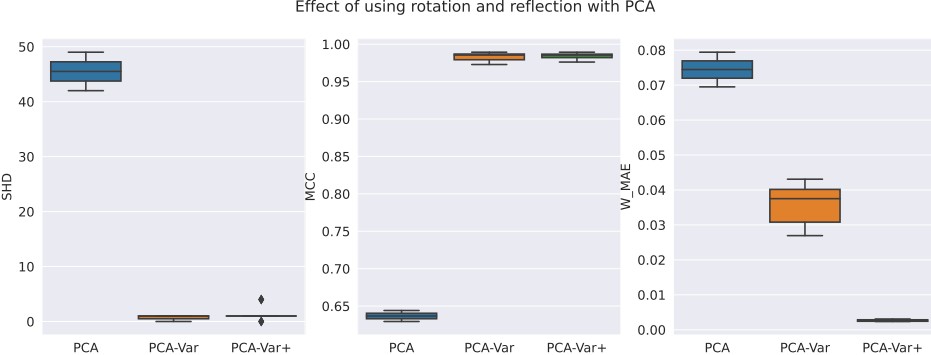

Figure 17: Comparison of 1) PCA without rotation (PCA), 2) PCA with the Varimax rotation (PCA-Var), and 3) PCA with the Varimax rotation and reflection (PCA-Var+). The Varimax rotation is crucial to recover the graph (left panel) and a good representation (middle panel). In order to recover the matrix W, the reflection operation is necessary (right panel).

