# OpenReview forum: "Causal Representation Learning in Temporal Data via Single-Parent Decoding"
_ICLR.cc/2024/Conference — ICLR 2024 Conference Withdrawn Submission_

### Official Review · Reviewer_d1tf · 2023-10-26

**Soundness:** 3 good
**Presentation:** 2 fair
**Contribution:** 2 fair
**Rating:** 3
**Confidence:** 4

**Summary:**

In this paper, the author proposes a causal discovery/representation method for temporal data. It assumes that each latent variable is a parent of one or more variables in the observational space. Given the specific structure, the author uses the orthogonal condition for regularization.

**Strengths:**

1. The organization of the paper is clear.

2. The author provides real-world experimental results on a specific domain (climate science).

**Weaknesses:**

1. My main concern is the technical contribution of this paper. The method presented in the paper is more like a practical solution for temporal data.  The assumption of single-parent is not surprising and limited to certain types of data, and the method used is not new as well ( the orthogonal regulation, the vae base model and etc).

2. The experimental verification is not very convincing. The author only compares with Varimax-PCMCI, DMS, and iVAE, and the results are only evaluated on MCC and SHD. None of these are SOTA methods. Then the author showcases the application in climate science, which could not support the proposed method is better than existing works.

**Questions:**

1. What is the benefit of learning the latent representation and the causal graph at the same time, compared to learning it separately?

2. Why does the single parent decoding provide better intuition on the latent space? It seems that the semantic meaning of the latent is still not available.

3. Why this method is a causal representation learning method?  The reason why I ask this question is that I do not see any evaluation or analysis from a causal perspective.

---

### Official Review · Reviewer_fLAr · 2023-10-30

**Soundness:** 3 good
**Presentation:** 2 fair
**Contribution:** 2 fair
**Rating:** 3
**Confidence:** 3

**Summary:**

The authors propose a causal representation learning method that can learn the latent variables and causal graphs over them on time series data under an assumption, namely single-parent decoding.

**Strengths:**

1. The proposed method can learn the nonlinearity in the relationships between latent variables and the mapping from latent to observational variables.
2. The authors prove the identifiability of both latent representation and its causal graph.
3. The authors conduct experiments on the simulated dataset, compared with some baseline algorithms.
4. The real-world experiment is interesting, and the dataset is high-dimensional, showing that the proposed algorithm works well with high-dimensional observations.

**Weaknesses:**

1. The single-parent decoding assumption seems to be a very strong assumption on time series data, given that time series data often exhibit autocorrelation, which is indeed a key characteristic of such data. To be more precise, there are typically causal relationships among observational data, resulting in multiple parents.
2. The paper lacks an explicit assumption section that systematically outlines all the assumptions required in the algorithm. Examples of such assumptions may include causal stationarity, the absence of instantaneous causal relationships for guaranteed identifiability, and others.
3. The content of experiment plots in Fig. 2-3 is not very informative as the various conditions are limited to just two or three distinct values.
4. The experimental results do not convincingly demonstrate that the proposed method has significant advantages over the baseline algorithm, as suggested in the main paper, where it states that "Varimax-PCMCI performs slightly better than CDSD in most conditions, except for more challenging cases...". The "more challenging cases" specifically involve only two synthetic datasets with time lags ($\tau = 3$) and dense graphs ($p = 0.3$). Would it be feasible to conduct additional experiments with larger values of $\tau$ and $p$ to gather more conclusive evidence? Furthermore, the paper lacks specific details in the plots for the nonlinear cases, which represent the advantageous scenarios for the proposed method.  For example, the paper only includes Fig.2b for nonlinear latent dynamics cases with various samples and only includes Fig.3b for nonlinear decoding cases with different densities $p$.
5. The number of baseline algorithms is limited. Could you please briefly explain why Mapped-PCMCI mentioned in the related work has not been applied as another baseline?

**Questions:**

1. In equation (3), the observational variable given its latent parent follows a normal distribution. Is this an assumption that was not explicitly mentioned in the paper?
2. In equation (4), what is the definition of $p(Z^{t}|Z^{<t})$ for $t\leq\tau$?
3. In the synthetic experiment section, is there any reason that the case of nonlinear dynamics and nonlinear decoding has not been discussed?
4. In Fig.2, $\tau$ varies from 1 to 3, and the SHD of the proposed method first increases and then decreases. Could you briefly explain why this happened? The relation between the performance of the method and $\tau$ is not apparent unless further investigation includes additional values of $\tau$, such as exploring how the SHD changes for $\tau=4, 5, \ldots$.
5. In Fig.2b, the time issue for Varimax-PCMCI is reasonable, however, is it feasible to conduct more experiments for $T<1000$? If the computation time for $T=5000$ is excessive, could experiments be performed for $T=1500$ or $T=2000$ for Varimax-PCMCI, thereby enabling a more comprehensive comparison?
6. When comparing the proposed method with iVAE and DMS, the constraint on $W$ is not applied to both baselines. How does this modification lead to a fairer comparison, given that the constraint ensures that the single-parent decoding assumption holds?
7. Is there any reason to apply linear dynamics and linear decoding in the real-world dataset, but not other versions?
8. In Fig.5a, does the statement "The groups are colored and numbered based on the latent variable to which they are related" suggest that the small circles sharing the same color correspond to the same latent parents? If so, given that the color of clusters 39, 41, 40 seems the same to me, does this mean they are clustered based on the same latent parents?
9. With $d_z=50$, could you briefly explain how to get the causal graph in  Fig.5c? Is it possible for the estimated causal graph to contain cycles? For instance, could it be the case that $Z_1^t \rightarrow Z_2^{t+1}$ and $Z_2^t \rightarrow Z_1^{t+1}$?

---

### Official Review · Reviewer_EMGp · 2023-11-01

**Soundness:** 3 good
**Presentation:** 3 good
**Contribution:** 3 good
**Rating:** 6
**Confidence:** 3

**Summary:**

This paper proposes Causal Discovery with Single-parent Decoding (CDSD) that could learn the underlying latent variables (high-level features) and a causal graph simultaneously from time-series data. The single-parent assumption they make helps the identifiability of both the latent representations and the causal graph. They run experiments on both a synthetic dataset as well as a real-world climate science dataset to show the applicability of their proposed methods.

**Strengths:**

1. Their proposed method pushes one step further in the challenging problem of causal representation learning.
2. The single-parent decoding assumption is very interesting and helps the  identifiability of both the latent representations and the causal graph
3. Their proposed method seems to be able to capture the correct latent representations and the causal link both in the synthetic dataset and the real-world climate science datasets, which seems to be really promising.
4. The paper is well-written and easy to follow. The authors included limitations of their work in the Discussion.

**Weaknesses:**

- CDSD performs worse in the linear case from Figure 2. I understand that the authors already claim they expect the advantage of CDSD under non-linear decoding (mappings from the observables to latents). But still, its worse performance under linear decoding is concerning.

**Questions:**

I have included some of the questions in the part of Weakness. Other than that:
1. In the paper, the connection between the latent variables, $G^k$, is set to be a binary matrix instead of a continuous one. Is it possible to extend to continuous case? And what are the pros and cons of different choices?
2. Can you discuss the validity of the conditional independence for the observables and latent variables respectively in practice?
3. Under linear setting, what is the reason to set the decoding function $r_j$ to be the identify function?
4. Is the paper using a different citation style?

---

### Official Review · Reviewer_bMUY · 2023-11-01

**Soundness:** 3 good
**Presentation:** 2 fair
**Contribution:** 2 fair
**Rating:** 3
**Confidence:** 3

**Summary:**

This paper proposes a differentiable causal discovery method for time series. The identifiability has been proved with the main assumption of single-parent decoding, which requires that each observed variable can only have one latent parent. Although the assumption might be helpful in specific application scenarios (as mentioned in the manuscript), it largely simplifies the identification task and thus the theoretical contribution is limited.

**Strengths:**

1. Real-world application to climate science has been conducted. The proposed method does find interesting results.

2. The implementation details have been clearly described. This part is very easy-to-follow.

**Weaknesses:**

1. The assumption of single-parent decoding may oversimplify the task. If one assumes that each observed variable can only have one latent variable as its parent, then it seems that there is no mixture at all. The manuscripts did mention a set of potential scenarios where that assumption could be helpful, but, still, the technical contributions seem to be limited because of the lack of mixture.

2. In the section of related work, the proposed method has been classified as causal representation learning with assumptions only on the map from latent variables to observations. However, this might be misleading since the method still requires time series data, of which the time index can be seen as a type of auxiliary variable. At the same time, the single-parent assumption seems to be much stronger than most of the existing sparsity assumptions. Together with the requirement of auxiliary variables (i.e., time indices), it seems that the considered problem setting could be probably solved by previous results?

3. Similarly, it is also not very reasonable to motivate the single-parent assumption by drawing its connection to independent mechanism analysis (i.e., orthogonal columns of the Jacobian). If each observed variable only has one parent, then the columns of the Jacobian actually have a disjoint set of indices of non-zero entries. This is a rather trivial case of orthogonality, and the connection based on that could lead to confusion

**Questions:**

Since I am mainly concerned about its theoretical contribution compared to previous results in causal representation learning, any further clarification would be much appreciated.